# ArCL: Enhancing Contrastive Learning with Augmentation-Robust Representations

**Xuyang Zhao**[1,2*] **Tianqi Du**[1,3*] **Yisen Wang**[3,4] **Jun Yao**[5] **Weiran Huang**[2†]

[1] School of Mathematical Sciences, Peking University

[2] Qing Yuan Research Institute, Shanghai Jiao Tong University

[3] National Key Lab of General Artificial Intelligence, School of Intelligence Science
and Technology, Peking University

[4] Institute for Artificial Intelligence, Peking University    [5] Huawei Noah's Ark Lab

## Abstract

Self-Supervised Learning (SSL) is a paradigm that leverages unlabeled data for model training. Empirical studies show that SSL can achieve promising performance in distribution shift scenarios, where the downstream and training distributions differ. However, the theoretical understanding of its transferability remains limited. In this paper, we develop a theoretical framework to analyze the transferability of self-supervised contrastive learning, by investigating the impact of data augmentation on it. Our results reveal that the downstream performance of contrastive learning depends largely on the choice of data augmentation. Moreover, we show that contrastive learning fails to learn domain-invariant features, which limits its transferability. Based on these theoretical insights, we propose a novel method called Augmentation-robust Contrastive Learning (ArCL), which guarantees to learn domain-invariant features and can be easily integrated with existing contrastive learning algorithms. We conduct experiments on several datasets and show that ArCL significantly improves the transferability of contrastive learning.

## 1 Introduction

A common assumption in designing machine learning algorithms is that training and test samples are drawn from the same distribution. However, this assumption may not hold in real-world applications, and algorithms may suffer from distribution shifts, where the training and test distributions differ. This issue has motivated a plethora of research in various settings, such as transfer learning, domain adaptation and domain generalization (Blanchard et al., 2011; Muandet et al., 2013; Wang et al., 2021a; Shen et al., 2021). Different ways of characterizing the relationship between test and training distributions lead to different algorithms. Most literature studies this in the supervised learning scenario. It aims to find features that capture some invariance across different distributions, and assume that such invariance also applies to test distributions (Peters et al., 2016; Rojas-Carulla et al., 2018; Arjovsky et al., 2019; Mahajan et al., 2021; Jin et al., 2020; Ye et al., 2021).

Self-Supervised Learning (SSL) has attracted great attention in many fields (He et al., 2020; Chen et al., 2020; Grill et al., 2020; Chen & He, 2021; Zbontar et al., 2021). It first learns a representation from a large amount of unlabeled training data, and then fine-tunes the learned encoder to obtain a final model on the downstream task. Due to its two-step nature, SSL is more likely to encounter the distribution shift issue. Exploring its transferability under distribution shifts has become an important topic. Some recent works study this issue empirically (Liu et al., 2021; Goyal et al., 2021; von Kügelgen et al., 2021; Wang et al., 2021b; Shi et al., 2022). However, the theoretical understanding is still limited, which also hinders the development of algorithms.

In this paper, we study the transferability of self-supervised contrastive learning in distribution shift scenarios from a theoretical perspective. In particular, we investigate which downstream distribu-

---

*Equal Contribution. This work was partially done when Xuyang was visiting Qing Yuan Research Institute.
†Correspondence to Weiran Huang (weiran.huang@outlook.com).

tions will result in good performance for the representation obtained by contrastive learning. We study this problem by deriving a connection between the contrastive loss and the downstream risk. Our main finding is that data augmentation is essential: contrastive learning *provably* performs well on downstream tasks whose distributions are close to the *augmented* training distribution.

Moreover, the idea behind contrastive learning is to find representations that are invariant under data augmentation. This is similar to the domain-invariance based supervised learning methods, since applying each kind of augmentation to the training data can be viewed as inducing a specific domain. Unfortunately, from this perspective, we discover that contrastive learning fails to produce a domain-invariant representation, limiting its transferability. To address this issue, we propose a new method called Augmentation-robust Contrastive Learning (ArCL), which can be integrated with various widely used contrastive learning algorithms, such as SimCLR (Chen et al., 2020) and MoCo (He et al., 2020). In contrast to the standard contrastive learning, ArCL forces the representation to align the two farthest positive samples, and thus provably learns domain-invariant representations. We conducted experiments on various downstream tasks to test the transferability of representations learned by ArCL on CIFAR10 and ImageNet. Our experiments demonstrate that ArCL significantly improves the standard contrastive learning algorithms.

### RELATED WORK

**Distribution shift in supervised learning.** Distribution shift problem has been studied in many literature (Blanchard et al., 2011; Muandet et al., 2013; Wang et al., 2021a; Shen et al., 2021). Most works aim to learn a representation that performs well on different source domains simultaneously (Rojas-Carulla et al., 2018; Mahajan et al., 2021; Jin et al., 2020), following the idea of causal invariance (Peters et al., 2016; Arjovsky et al., 2019). Structural equation models are often assumed for theoretical analysis (von Kügelgen et al., 2021; Liu et al., 2020; Mahajan et al., 2021). Distributionally robust optimization optimizes a model's worst-case performance over some uncertainty set directly (Krueger et al., 2021; Sagawa et al., 2019; Duchi & Namkoong, 2021; Duchi et al., 2021). Stable learning (Shen et al., 2020; Kuang et al., 2020) learns a set of global sample weights that could remove the confounding bias for all the potential treatments from data distribution. Disentangled representation learning (Bengio et al., 2013; Träuble et al., 2021; Kim & Mnih, 2018) aims to learn representations where distinct and informative factors of variations in data are separated.

**Theoretical understanding of contrastive learning.** A number of recent works also aim to theoretically explain the success of contrastive learning in IID settings. One way to explain it is through the mutual information between positive samples (Tian et al., 2020; Hjelm et al., 2018; Tschannen et al., 2019). Arora et al. (2019) directly analyze the generalization of InfoNCE loss based on the assumption that positive samples are drawn from the same latent classes. In the same setting, Bao et al. (2022) establish equivalence between InfoNCE and supervised loss and give sharper upper and lower bounds. Huang et al. (2021) take data augmentation into account and provide generalization bounds based on the nearest neighbor classifier.

**Contrastive learning in distribution shift.** Shen et al. (2022) and HaoChen et al. (2022) study contrastive learning in unsupervised domain adaptation, where unlabeled target data are obtained. Shi et al. (2022) show that SSL is the most robust on distribution shift datasets compared to autoencoders and supervised learning. Hu et al. (2022) improve the out-of-distribution performance of SSL from an SNE perspective. Other robust contrastive learning methods (Kim et al., 2020; Jiang et al., 2020) focus on adversarial robustness while this paper focuses on distributional robustness.

## 2 PROBLEM FORMULATION

Given a set of unlabeled data where each sample $X$ is i.i.d. sampled from training data distribution $\mathcal{D}$ on $\mathcal{X} \subseteq \mathbb{R}^d$, the goal of Self-Supervised Learning (SSL) is to learn an encoder $f \colon \mathcal{X} \to \mathbb{R}^m$ for different downstream tasks. Contrastive learning is a popular approach of SSL, which augments each sample $X$ twice to obtain a positive pair $(X_1, X_2)$, and then learns the encoder $f$ by pulling them close and pushing random samples (also called negative samples) away in the embedding space. The data augmentation is done by applying a transformation $A$ to the original data, where $A$ is randomly selected from a transformation set $\mathcal{A}$ according to some distribution $\pi$. We use

$\mathcal{D}_A$ to denote the distribution of augmented data under a specific transformation $A$, and use $\mathcal{D}_\pi$ to denote the distribution of augmented data, i.e., $\mathcal{D}_\pi = \int_{\mathcal{A}} \mathcal{D}_A \, d\pi(A)$. The loss of contrastive learning typically consists of two terms:

$$\mathcal{L}_{\text{con}}(f; \mathcal{D}, \pi) := \mathcal{L}_{\text{align}}(f; \mathcal{D}, \pi) + \lambda \mathcal{L}_{\text{reg}}(f; \mathcal{D}, \pi), \tag{1}$$

where $\mathcal{L}_{\text{align}}(f; \mathcal{D}, \pi) := \mathbb{E}_{X \sim \mathcal{D}} \mathbb{E}_{(A_1, A_2) \sim \pi^2} \|f(A_1(X)) - f(A_2(X))\|^2$ measures the alignment of positive samples[1], and $\mathcal{L}_{\text{reg}}(f; \mathcal{D}, \pi)$ is the regularization term to avoid feature collapse. For example, the regularization term in InfoNCE loss is

$$\mathcal{L}_{\text{reg}}(f; \mathcal{D}, \pi) := \mathbb{E}_{(X, X') \sim \mathcal{D}^2} \mathbb{E}_{(A_1, A_2, A') \sim \pi^3} \log \left( e^{f(A_1(X))^\top f(A_2(X))} + e^{f(A_1(X))^\top f(A'(X'))} \right).$$

In this paper, we consider multi-class classification problems $\mathcal{X} \subseteq \mathbb{R}^d \to \mathcal{Y} = \{1, \ldots, K\}$ as downstream tasks, and focus on the covariate shift setting, i.e., target distribution $\mathcal{D}^{\text{tar}}$ is different from $\mathcal{D}$ but $\mathbb{P}(Y|X)$ is fixed. We use the best linear classifier on top of $f$ for downstream classification, and evaluate the performance of $f$ on $\mathcal{D}^{\text{tar}}$ by its risk (also used by Shi et al. (2022)):

$$\mathcal{R}(f; \mathcal{D}^{\text{tar}}) := \min_{h \in \mathbb{R}^{K \times m}} \mathbb{E}_{X \sim \mathcal{D}^{\text{tar}}} \ell(h \circ f(X), Y), \tag{2}$$

where $\ell$ is the loss function. We focus on the case where $\ell$ is the square loss. With a slight abuse of notation, for the classifier $h \circ f : \mathbb{R}^m \to \mathbb{R}^k$, we also use $\mathcal{R}(h \circ f; \mathcal{D}^{\text{tar}})$ to denote its risk. Our goal is to study the transferability of representation learned by contrastive learning on different downstream distributions. For simplicity, we assume that $\|f(x)\| = 1$ for every $x \in \mathcal{X}$ and $f$ is $L$-Lipschitz continuous throughout this paper.

At the end of this section, we remark the difference between transformations and augmentations. Transformations are deterministic mapping from $\mathcal{X}$ to $\mathcal{X}$, and augmentations are random application of them. We give examples to demonstrate that many augmentation methods used in practice indeed satisfy this model.

**Example 2.1.** *The augmentations used in SimCLR (Chen et al., 2020) are compositions of random transformations, such as RandomCrop, HorizonalFlip and Color distortion. Each $A$ denotes a specific composition here. Users will determine probabilities that each transformation will be applied. If used, the parameters of the random transformation such as crop range are also selected randomly. Until the parameters of each used transformations are set, the composited transformation is deterministic. Therefore, we can view these augmentation processes as randomly selecting some deterministic transformations, following the distribution $\pi$ determined by users.*

**Example 2.2.** *Some recent works (Jahanian et al., 2021) propose to augment data by transformations in the latent space. Suppose we have a pre-trained generative model with an encoder $T : \mathcal{X} \to \mathcal{Z}$ and a generator $G : \mathcal{Z} \to \mathcal{X}$, where $\mathcal{Z}$ is a latent space. Since $\mathcal{Z}$ is usually simple, one can augment data in $\mathcal{X}$ by randomly shifting $T(X)$. In this setting, the augmentation can be parameterized by $A_\theta(X) = G(T(X) + \theta)$ for $\theta \in \mathcal{Z}$. The distribution $\pi$ of $\theta$ is usually chosen to be a normal distribution.*

## 3 TRANSFERABILITY ANALYSIS: THE CRUCIAL ROLE OF DATA AUGMENTATION

In this section, we theoretically demonstrate that data augmentation plays a crucial role in the transferability of contrastive learning.

The first step is to establish the connection between contrastive loss $\mathcal{L}_{\text{con}}(f; \mathcal{D}, \pi)$ and downstream risk $\mathcal{R}(f; \mathcal{D}^{\text{tar}})$. For this purpose, we adopt the $(\sigma, \delta)$-augmentation notion proposed by Huang et al. (2021), which was used there to study the KNN error of contrastive learning.

**Definition 3.1** ($(\sigma, \delta)$-transformation). *Let $C_k \subseteq \mathcal{X}$ be the set of all points in class $k$. A data transformation set $\mathcal{A}$ is called a $(\sigma, \delta)$-transformation on $\mathcal{D}$ for some $\sigma \in (0, 1]$ and $\delta > 0$, if for every $k \in [K]$, there exists $C_k^0 \subseteq C_k$ such that*

$$\mathbb{P}_{X \sim \mathcal{D}}(X \in C_k^0) \geq \sigma \, \mathbb{P}_{X \sim \mathcal{D}}(X \in C_k), \text{ and } \sup_{x_1, x_2 \in C_k^0} d_{\mathcal{A}}(x_1, x_2) \leq \delta,$$

*where $d_{\mathcal{A}}(x_1, x_2) := \inf_{A_1, A_2 \in \mathcal{A}} d(A_1(x_1), A_2(x_2))$ for some distance $d(\cdot, \cdot)$.*

---

[1]In this paper, notation $\|\cdot\|$ stands for $L^2$-norm or Frobenius norm for vectors and matrices, respectively.

This definition measures the concentration of augmented data. A transformation set with smaller $\delta$ and larger $\sigma$ clusters original data more, i.e., samples from the same class are closer after augmentation. Therefore, one can expect the learned representation $f$ to have better cluster performance, which was measured by KNN error in (Huang et al., 2021). We remark that most of contrastive learning theories (Arora et al., 2019; Ash et al., 2021; Bao et al., 2022) assume that given class $k$, positive pairs are obtained by i.i.d. sampling from $\mathbb{P}_{X \sim \mathcal{D}}(\cdot \mid X \in C_k)$. Compared with this data generation model, our $(\sigma, \delta)$ notion is more practical.

By taking into account the linear classifier on top of $f$, we extend the KNN result in (Huang et al., 2021) to the downstream risk, and obtain the following lemma.

**Lemma 3.1.** *For any distribution $\mathcal{D}$ and encoder $f$, define $\mu_k(f; \mathcal{D}) := \mathbb{E}_{X \sim \mathcal{D}} f(X)$ for $k \in [K]$. Suppose that $\mathcal{A}$ is a $(\sigma, \delta)$-transformation. For any representation $f : \mathcal{X} \to \mathbb{R}^{d_1}$ and linear layer $h \in R^{K \times d_1}$, we have*

$$\mathcal{R}(h \circ f; \mathcal{D}_\pi) \leq c\|h\|\sqrt{K}\sigma(\mathcal{L}_{\text{align}}(f; \mathcal{D}, \pi))^{\frac{1}{4}} + \|h\|\tau(\sigma, \delta) + \sum_{k=1}^{K} \mathcal{D}_\pi(C_k)\|e_k - h \circ \mu_k(f; \mathcal{D}_\pi)\|, \quad (3)$$

*where $c$ is an absolute constant and $\tau(\sigma, \delta)$ is a constant which only depends on $(\sigma, \delta)$. The detail form of $\tau$ is shown in the appendix.*

The first term in equation 3 is the alignment of $f$, which is optimized during the contrastive learning pre-training on $\mathcal{D}$. The second term is determined solely by the $(\sigma, \delta)$ quantity of the data augmentation. Larger $\sigma$ and smaller $\delta$ induce smaller $\tau(\sigma, \delta)$. These two terms together quantify the cluster level of the representation $f$ learned by contrastive learning. The third term is related to the linear layer $h$ on top of $f$, and will be minimized in downstream training, where each class center is converted to the corresponding ground-truth label. If the regularization term $\mathcal{L}_{\text{reg}}$ is appropriately chosen, the class centers can be distinguished from each other, and the third term can be reduced to $0$ by $h$.

Taking the linear classifier into account, we obtain the following main result.

**Theorem 3.2.** *Suppose that $\mathcal{A}$ is a $(\sigma, \delta)$-transformation on $\mathcal{D}$, and $\pi$ is the augmentation distribution on $\mathcal{A}$. For an encoder $f$, define*

$$\gamma_{\text{reg}}(f; \mathcal{D}, \pi) = 1 - c_1(\mathcal{L}_{\text{align}}(f; \mathcal{D}, \pi))^{1/4} - \tau - c_2 \max_{k \neq k'} |\mu_k(f; \mathcal{D}_\pi)^\top \mu_{k'}(f; \mathcal{D}_\pi)|$$

*for some positive constants $c_1$ and $c_2$. Then for any $f$ such that $\{\mu_k(f; \mathcal{D}_\pi)\}_{k=1}^K$ are linearly independent and $\gamma_{reg}(f; \mathcal{D}, \pi) > 0$, we have*

$$\mathcal{R}(f; \mathcal{D}_\pi) \leq c \cdot \left( \gamma_{\text{reg}}^{-1}(f; \mathcal{D}, \pi) \left( \mathcal{L}_{\text{align}}(f; \mathcal{D}, \pi) \right)^{1/4} + \gamma_{\text{reg}}^{-1}(f; \mathcal{D}, \pi)\tau(\sigma, \delta) \right), \quad (4)$$

*where $c$ is an absolute constant.*

**Remark 1.** *(a) The linear independence of $\{\mu_k(f; \mathcal{D}_\pi)\}_{k=1}^K$ condition is weak, since its complement event is a null set in Euclidean space under Lebesgue measure, and have $0$ probability under any distribution that is absolutely continuous w.r.t. Lebesgue measure.*

*(b) The quantity $\gamma_{\text{reg}}^{-1}$ is essentially the bound of $L_2$ norm of the linear classifier $h$ which reduces the third term $\sum_{k=1}^K \mathcal{D}_\pi(C_k)\|e_k - h \circ \mu_k(f; \mathcal{D}_\pi)\|$ in equation 3 to $0$, i.e., mapping the class centers to their corresponding ground-truth labels. The more orthogonal the class centers, the larger the $\gamma_{\text{reg}}$ and the better the bounds. The theory developed in (Huang et al., 2021) shows that popular algorithms such as SimCLR and Barlow Twins can indeed push away the class centers, and the constant $\gamma_{\text{reg}}$ can be ensured to have a constant gap away from $0$. Therefore, for simplicity we directly impose this requirement on $f$.*

This theorem shows that contrastive learning on distribution $\mathcal{D}$ with augmentation $\pi$ is essentially optimizing the supervised risk on the augmented distribution $\mathcal{D}_\pi$, instead of the original distribution $\mathcal{D}$. Therefore, the augmentation $\pi$ is crucial to the transferability of contrastive learning. If the downstream distribution $\mathcal{D}^{\text{tar}}$ is close to $\mathcal{D}_\pi$, the encoder obtained by contrastive learning performs well on it. The more diverse $\pi$ is, we can expect $\mathcal{D}_\pi$ to approximate downstream distributions more accurately, and contrastive learning has better transferability.

Besides, if we have some prior knowledge of downstream distribution, we can specify an appropriate augmentation such that $\mathcal{D}_\pi$ and $\mathcal{D}^{\text{tar}}$ are close in order to improve downstream performance. For example, consider the case where original data $\mathcal{D}$ are gray digits, and the downstream distribution $\mathcal{D}^{\text{tar}}$ is obtained by dying them red. Then if we use random coloring as data augmentation and conduct contrastive learning on $\mathcal{D}$, the learned feature will generalize to $\mathcal{D}^{\text{tar}}$ well, even though $\mathcal{D}^{\text{tar}}$ is far away from $\mathcal{D}$.

## 4 TRANSFERABILITY LIMITATION: A DOMAIN-INVARIANCE VIEW

Our theoretical results in the last section suggests that data augmentation is crucial to the transferability of contrastive learning. However, there exists a fundamental limitation of contrastive learning: *the learned representation is not domain-invariant,* which is important for a model to generalize to more downstream datasets (Peters et al., 2016; Arjovsky et al., 2019).

Roughly speaking, domain-invariance means that the learned model can extract some intrinsic features, which are invariant across different domains. In supervised domain generalization, a common setting is to have a training domain set consisting of multiple source domains. In contrastive learning, correspondingly, we can also consider $\mathcal{D}_A$ as a *transformation-induced domain* for each data transformation $A$, and consider $\{\mathcal{D}_A\}_{A \in \mathcal{A}}$ as the training domain set. Since the goal of contrastive learning is also to align different $\mathcal{D}_A$, we naturally expect that the features it learn are domain-invariant. However, since the alignment loss $\mathcal{L}_{\text{align}}$ in contrastive learning is obtained by averaging over different transformations (i.e., taking expectation), the learned features are not invariant even across $\{\mathcal{D}_A\}_{A \in \mathcal{A}}$. In other words, encoders learned through contrastive learning could behave extremely differently in different $\mathcal{D}_A$.

The following toy model gives an illustrative example.

**Proposition 4.1.** *Consider a two-dimensional classification problem with data* $(X_1, X_2) \sim \mathcal{N}(0, I_2)$. *The label $Y$ satisfies $Y = 1(X_1 \geq 0)$, and the data augmentation is to multiply $X_2$ by standard normal noise, i.e.,*

$$A_\theta(X) = (X_1, \theta \cdot X_2),$$
$$\theta \sim \mathcal{N}(0, 1).$$

*The corresponding transformation-induced domain set is $\mathcal{P} = \{\mathcal{D}_c : \mathcal{D}_c = (X_1, c \cdot X_2) \text{ for } c \in \mathbb{R}\}$. We consider the 0-1 loss in equation 2. Then for every $\varepsilon > 0$, there exists representation $f$ and two domains $\mathcal{D}_c$ and $\mathcal{D}_{c'}$ such that*

$$\mathcal{L}_{\text{align}}(f; \mathcal{D}, \pi) < \varepsilon,$$

*but*

$$|\mathcal{R}(f; \mathcal{D}_c) - \mathcal{R}(f; \mathcal{D}_{c'})| \geq \frac{1}{4}$$

This example shows that there exists a representation with arbitrary small contrastive loss, but with very different performance over different transformation-induced domains. The intuition behind this example is that in order to make $\mathcal{L}_{\text{align}}$ small, it is sufficient to align different domains (induced by different transformations) in an average rather than uniform sense. Therefore, the representation may still suffer a large alignment loss on some rarely selected augmented domains.

## 5 ARCL: AUGMENTATION-ROBUST CONTRASTIVE LEARNING

To further improve the transferability performance of contrastive learning, we consider how to learn a representation that is domain-invariant across $\{\mathcal{D}_A\}_{A \in \mathcal{A}}$. First of all, we need a formal definition that mathematically characterize domain-invariance, so that we can design algorithms based on it. Here, we borrow the notion proposed by (Arjovsky et al., 2019).

**Definition 5.1.** *We say that a representation $f$ elicits an invariant predictor $h_0 \circ f$ across a domain set $\mathcal{P}$, if there is a classifier $h_0$ simultaneously optimal for all domains in $\mathcal{P}$, that is, $h_0 \in \arg\min_h \mathcal{R}(h \circ f; \mathcal{D})$ for all $\mathcal{D} \in \mathcal{P}$.*

This definition is equivalent to learning features whose correlations with the target variable are stable, and has been shown to improve distribution shift transferability both practically and theoretically in supervised learning. Interestingly, by setting $\mathcal{P}$ to be $\{\mathcal{D}_A\}_{A \in \mathcal{A}}$, this definition is well

suited to contrastive learning. Specifically, define the *augmentation-robust loss* as

$$\mathcal{L}_{\mathrm{AR}}(f;\mathcal{D}) := \mathop{\mathbb{E}}_{X \in \mathcal{D}} \sup_{A,A' \in \mathcal{A}} \|f(A(X)) - f(A'(X))\|^2,$$

which is an uniform version of the original alignment loss $\mathcal{L}_{\mathrm{align}}$. Then we have the following results.

**Theorem 5.1.** *For any two transformations $A$ and $A'$, linear predictor $h$ and representation $f$, we have*

$$\sup_{A,A' \in \mathcal{A}} |\mathcal{R}(h \circ f; \mathcal{D}_A) - \mathcal{R}(h \circ f; \mathcal{D}_{A'})| \le c \cdot \|h\| L_{\mathrm{AR}}(f, \mathcal{D}). \tag{5}$$

*Moreover, fix $f$ and let $h_A \in \arg\min_h \mathcal{R}(h \circ f, \mathcal{D}_A)$. Then we have*

$$|\mathcal{R}(h_A \circ f; \mathcal{D}_{A'}) - \mathcal{R}(h_{A'} \circ f; \mathcal{D}_{A'})| \le 2c \cdot (\|h_A\| + \|h_{A'}\|) L_{\mathrm{AR}}(f, \mathcal{D}). \tag{6}$$

Compared with $\mathcal{L}_{\mathrm{align}}$, $\mathcal{L}_{\mathrm{AR}}$ replaces the expectation over $\mathcal{A}$ by the supremum operator, hence $\mathcal{L}_{\mathrm{align}}(f; \mathcal{D}, \pi) \le \mathcal{L}_{AR}(f; \mathcal{D})$ for all $f$ and $\pi$. If $\mathcal{L}_{AR}(f)$ is small, the above proposition shows that $\mathcal{R}(h \circ f; \mathcal{D}_A)$ varies slightly with $A$, so that the optimal $h$ for $\mathcal{D}_A$ is close to that for $\mathcal{D}_{A'}$. In other words, representation with smaller $\mathcal{L}_{AR}$ tends to *elicit the same linear optimal predictors* across different domains, a property that does not hold for original alignment loss. Small $L_{\mathrm{AR}}$ not only requires $f$ to push the positive pairs closer, but also forces this alignment to be uniform across all transformations in $\mathcal{A}$.

Therefore, we propose to train the encoder using the augmentation-robust loss for better distribution shift transfer performance, i.e., replace $\mathcal{L}_{\mathrm{align}}$ by $\mathcal{L}_{AR}$ in contrastive loss equation 1. In practical implementation, $\sup_{A_1, A_2 \in \mathcal{A}} \|f(A_1(X)) - f(A_2(X))\|_2^2$ can not be computed exactly, unless $|\mathcal{A}|$ is finite and small. Therefore, we use the following approach to approximate it. For each sample $X$, we first randomly select $m$ augmentations to obtain a set of augmented samples denoted by $\widehat{\mathcal{A}}_m(X)$. Then, rather than taking average as in standard contrastive learning, we consider use

$$\sup_{A_1, A_2 \in \widehat{\mathcal{A}}_m(X)} \|f(A_1(X)) - f(A_2(X))\|_2^2$$

as the alignment term, which is an approximation of $\sup_{A_1, A_2 \in \mathcal{A}} \|f(A_1(X)) - f(A_2(X))\|_2^2$. The integer $m$ is called the *number of views*, which controls the approximation accuracy. Then, given $n$ unlabeled samples $(X_1, \ldots, X_n)$, the empirical augmentation-robust loss is

$$\widehat{\mathcal{L}}_{\mathrm{AR}}(f) := \frac{1}{n} \sum_{i=1}^{n} \sup_{A_1, A_2 \in \widehat{\mathcal{A}}_m(X_i)} \|f(A_1(X_i)) - f(A_2(X_i))\|_2^2. \tag{7}$$

We remark that our proposed method can be applied to any contrastive learning methods whose objective can be formulated by equation 1, such as SimCLR, Moco, Barlow Twins (Zbontar et al., 2021) and so on. We give the detailed form of SimCLR + ArCL as an example in Algorithm 1, and provide MoCo + ArCL in the Appendix. The difference between Algorithm 1 and SimCLR is only the construction of positive pairs. In each epoch $t$, for every sample $X$, we first randomly select $m$ transformations denoted by $\widehat{\mathcal{A}}(X) = \{A_1, \ldots, A_m\}$. Then based on current encoder $f$ and projector $g$, we select two transformations that have worst alignment (minimal inner product) among all pairs in $\widehat{\mathcal{A}}(X)$, and use them to construct the finally used positive pairs of $X$. The construction of negative pairs and update rule are the same as SimCLR.

## 5.1 THEORETICAL JUSTIFICATION

We now give theoretical guarantees of our proposed approximation, i.e., bound the gap between $\widehat{\mathcal{L}}_{\mathrm{AR}}(f)$ and $\mathcal{L}_{\mathrm{AR}}(f)$. For each representation $f \in \mathcal{F}$, where $\mathcal{F}$ is some hypothesis space, define $z_f(x) := \sup_{A_1, A_2 \in \mathcal{A}} \|f(A_1 x) - f(A_2 x)\|_2$ and $\mathcal{Z}_{\mathcal{F}} := \{z_f : f \in \mathcal{F}\}$. Then we use the Rademacher complexity of $\mathcal{Z}$ to define the *alignment complexity* of $\mathcal{F}$ as

$$\widetilde{\mathfrak{R}}_n(\mathcal{F}) := \mathfrak{R}_n(\mathcal{Z}_{\mathcal{F}}) = \mathbb{E}\left[\sup_{z \in \mathcal{Z}} \left(\frac{1}{n}\sum_{i=1}^{n} \sigma_i z(X_i)\right)\right].$$

This quantity is a characteristic of the complexity of $\mathcal{F}$ *in terms of its alignment*. If most of the functions in $\mathcal{F}$ have good alignment under $\mathcal{A}$, then $\widetilde{\mathfrak{R}}_n(\mathcal{F})$ is small even though $\mathfrak{R}_n(\mathcal{F})$ could be large. Using this definition, we have the following result.

---

**Algorithm 1:** SimCLR + ArCL

    **input** : Batch size $N$, temperature $\tau$, augmentation $\pi$, number of views $m$, epoch $T$,
             encoder $f$, projector $g$.

1   **for** $t = 1, \ldots, T$ **do**
2      sample minibatch $\{X_i\}_{i=1}^{N}$;
3      **for** $i = 1, \ldots, N$ **do**
4          draw $m$ augmentations $\widehat{\mathcal{A}} = \{A_1, \ldots, A_m\} \sim \pi$;
5          $z_{i,j} = g(f(A_j X_i))$ for $j \in [m]$;
6          *# select the worst positive samples*;
7          $s_i^+ = \min_{j,k \in [m]}\{z_{i,j}^\top z_{i,k} / (\|z_{i,j}\| \|z_{i,k}\|)\}$;
8          *# select the negative samples*;
9          **for** $j = 1, \ldots, N$ **do**
10              $s_{i,j}^- = z_{i,1}^\top z_{j,1} / (\|z_{i,1}\| \|z_{j,1}\|)$;
11              $s_{i,j+N}^- = z_{i,1}^\top z_{j,2} / (\|z_{i,1}\| \|z_{j,2}\|)$ ;
12      compute $L = -\frac{1}{N} \sum_{i=1}^{N} \log \frac{\exp(s_i^+/\tau)}{\sum_{j=1, j \neq i}^{2N} \exp(s_{i,j}^-/\tau)}$;
13      update $f$ and $g$ to minimize $L$;
14 **return** $f$

---

**Theorem 5.2.** *Suppose that $\Theta = B_{d_\mathcal{A}}(1)$, i.e., the $d_\mathcal{A}$-dimensional unit ball. Suppose that all $A$ in $\mathcal{A}$ are $\mathcal{L}_\mathcal{A}$-Lipschitz continuous. Let $\pi$ be a distribution on $\Theta$ with density $p_\pi$ such that $\inf_{\theta \in \Theta} p_\pi(\theta) \geq c_\pi$. Then with probability at least $1 - 2\varepsilon$, for every $f \in \mathcal{F}$,*

$$\mathcal{L}_{\mathrm{AR}}(f) \leq \widehat{\mathcal{L}}_{\mathrm{AR}}(f) + 2\widetilde{\mathfrak{R}}_n(\mathcal{F}) + \left(\frac{\log(1/\varepsilon)}{n}\right)^{\frac{1}{2}} + LL_A \left(\frac{\log(2n/\varepsilon)}{c_\pi m}\right)^{\frac{1}{d_\mathcal{A}}},$$

*where $n$ is the training sample size and $m$ is the number of views.*

The bound in Theorem 5.2 consists of two terms $O(n^{-1/2})$ and $O(m^{-1/d_\mathcal{A}})$. The first one is due to finite sample issue, and the second is caused by the finite maximization approximation of the supremum. $1/c_\pi$ acts like the "volume" of the transformation set and is of constant order. As $m$ increases, the second term decreases and the bound becomes tighter. Note that we need to assume that the augmentation distribution $\pi$ behaves uniformly, i.e., $\inf_{\theta \in \Theta} p_\pi(\theta) \geq c$ for some positive constant $c$, otherwise we can not ensure that the finite maximization is an acceptable approximation.

## 6   EXPERIMENTS

We conduct experiments on several distribution shift settings to show the performance of our proposed methods. To show the adaptability of our approach, we apply ArCL on SimCLR (Chen et al., 2020) and MoCo V2 (He et al., 2020) respectively. Following the setting of SimCLR and MoCo V2, we add a projection head after the backbone in the pre-training stage and remove it in the downstream task. For the linear evaluation, we follow the protocol that the encoder is fixed and a linear classifier on the top of the encoder is trained using test set. For the full fine-tuning, we also add a linear classifier on the top of the encoder but we do not fix the encoder.

### 6.1   EXPERIMENTS ON MODIFIED CIFAR10 AND CIFAR100

**Setup.** Our representations are trained on CIFAR-10 (Krizhevsky, 2009) using SimCLR modified with our proposed ArCL. Different batch sizes $(256, 512)$ and number of views $(m = 4, 6, 8)$ are considered. We use ResNet-18 as the encoder and a 2-layer MLP as the projection head. We train the representation with 500 epochs. The temperature is $0.5$. Warm-up is not used. We conduct linear evaluation with 100 epochs on different downstream datasets to test the OOD performance of the learned representations.

Table 1: 5 different augmentations.

|       | Grayscale | RandomCrop | HorizontalFlip | ColorJitter |
|-------|:---------:|:----------:|:--------------:|:-----------:|
| Aug 1 | ✓ | – | – | – |
| Aug 2 | – | ✓ | – | – |
| Aug 3 | – | – | ✓ | – |
| Aug 4 | – | – | – | ✓ |
| Aug 5 | ✓ | – | – | ✓ |

Table 2: Linear evaluation results (%) of pretrained CIFAR10 models on CIFAR10, CIFAR100 and their modified versions.

|          | Method | Batch Size | Aug 1 | Aug 2 | Aug 3 | Aug 4 | Aug 5 | Original |
|----------|--------|:----------:|:-----:|:-----:|:-----:|:-----:|:-----:|:--------:|
| CIFAR10  | SimCLR | 256 | 86.36 | 83.21 | 86.93 | 86.42 | 86.13 | 86.76 |
|          | SimCLR + ArCL (views=4) | 256 | 88.68 | 86.77 | 89.01 | 88.70 | 88.31 | 88.95 |
|          | SimCLR + ArCL (views=6) | 256 | **88.95** | **87.18** | **89.54** | **88.92** | **88.61** | **89.11** |
|          | SimCLR | 512 | 88.62 | 86.27 | 88.96 | 88.56 | 88.37 | 88.81 |
|          | SimCLR + ArCL (views=4) | 512 | 89.97 | 88.06 | 90.48 | 89.91 | 89.59 | 90.20 |
|          | SimCLR + ArCL (views=6) | 512 | 90.24 | 89.54 | 90.69 | 90.43 | 90.07 | 90.69 |
|          | SimCLR + ArCL (views=8) | 512 | **90.44** | **88.96** | **90.98** | **90.63** | **90.31** | **90.84** |
| CIFAR100 | SimCLR | 256 | 51.65 | 47.55 | 53.17 | 52.05 | 51.36 | 52.75 |
|          | SimCLR+ArCL(views=4) | 256 | 53.76 | 49.80 | 55.68 | 54.19 | 52.96 | 54.83 |
|          | SimCLR+ArCL(views=6) | 256 | **54.13** | **50.74** | **55.74** | **54.75** | **53.46** | **55.29** |
|          | SimCLR | 512 | 52.28 | 48.09 | 53.45 | 52.58 | 51.53 | 53.12 |
|          | SimCLR+ArCL(views=4) | 512 | 53.40 | 50.16 | 54.92 | 53.77 | 52.61 | 54.20 |
|          | SimCLR+ArCL(views=6) | 512 | 54.00 | 50.57 | **56.24** | **55.04** | **53.77** | 55.60 |
|          | SimCLR+ArCL(views=8) | 512 | **54.59** | **50.85** | 55.74 | 54.62 | 53.21 | **55.96** |

**Datasets.** We use CIFAR10 to train the representation. The data augmentation used follows the setting in SimCLR, i.e., a composition of RandomCrop, RandomHorizontalFlip, ColorJitter and Random Grayscale. To evaluate the transferability of the learned representation, we use different downstream datasets to train the linear classifier and test its accuracy. Downstream datasets are generated by modifying the original CIFAR10 and CIFAR100 datasets, i.e., applying different data augmentations to them. We choose 5 different augmentations as shown in Table 1. The results are shown in Table 2.

**Results.** In all settings, our proposed approach improves the transferability of SimCLR significantly. As the number of views $m$ increases, the accuracy also increases, which matches our theoretical results. Besides, the performance improvement brought by increasing $m$ tends to saturate, which suggest that $m$ does not need to be very large.

## 6.2 EXPERIMENTS ON IMAGENET TRANSFERRING TO SMALL DATASETS

**Setup.** In this part, we will first train an encoder on ImageNet (Deng et al., 2009) using MoCo v2 modified with our proposed ArCL. We select the number of ArCL views from $\{2, 3, 4\}$. Notice that since there exists asymmetric network architecture in MoCo, there is difference between MoCo and MoCo + ArCL(views=2). We use ResNet50 as the backbone and follow all the other architecture settings of MoCo. The memory bank size $K$ is set to 65536 and the batch size is set to 256. For the training epochs, we propose three schemes: 1) 200 epochs training from scratch, 2) 50 epochs training from the model which is pretrained by MoCo for 800 epochs, 3) 100 epochs training from the model which is pretrained by MoCo for 800 epochs. For each setting, we search the best start learning rate in $\{0.03, 0.0075, 0.0015\}$ and the best temperature in $\{0.20, 0.15, 0.12, 0.10, 0.05\}$. We use the SGD optimizer and apply the cosine learning rate decay.

After the model has been pre-trained by contrastive learning, we test its transferability with linear evaluation and full fine-tuning on various downstream small datasets, following the settings in (Ericsson et al., 2021). For the target datasets, we adopt FGVC Aircraft (Maji et al., 2013), Caltech-101 (Fei-Fei et al., 2004), Stanford Cars (Krause et al., 2013), CIFAR10, CIFAR100, DTD (Cimpoi et al., 2014), Oxford 102 Flowers (Nilsback & Zisserman, 2008), Food-101 (Bossard et al., 2014) and Oxford-IIIT Pets (Parkhi et al., 2012). For linear evaluation, multinomial logistic regression is

Table 3: Results comparison on linear evaluation (up) and finetuning (down) of pretrained ImageNet models on popular recognition datasets. Additionally, we provide a supervised baseline for comparison, a standard pretrained ResNet50 available from the PyTorch library. The supervised baseline results are from (Ericsson et al., 2021). Results style: **best** in the same epoch setting. **Avg** takes the average of the results on all the nine small datasets.

| | | Epochs | ImageNet | Aircraft | Caltech101 | Cars | CIFAR10 | CIFAR100 | DTD | Flowers | Food | Pets | Avg |
|---|---|---|---|---|---|---|---|---|---|---|---|---|---|
| Linear | MoCo | 800 | 70.68 | 41.79 | 87.92 | 39.31 | 92.28 | 74.90 | 73.88 | 90.07 | 68.95 | 83.30 | 72.49 |
| | MoCo | 800+50 | 70.64 | 41.00 | 87.63 | 39.01 | 92.27 | 75.14 | 74.31 | 88.31 | 68.57 | 83.69 | 72.21 |
| | MoCo + ArCL(views=2) | 800+50 | 69.70 | 44.29 | 89.79 | 42.15 | 93.07 | 76.70 | 74.20 | 90.40 | 70.94 | 83.68 | 73.91 |
| | MoCo + ArCL(views=3) | 800+50 | 69.80 | 44.57 | 89.48 | 42.11 | **93.29** | **77.33** | 74.63 | 91.13 | 71.16 | **84.23** | 74.21 |
| | MoCo + ArCL(views=4) | 800+50 | 69.80 | **44.62** | **89.66** | **42.88** | 93.22 | 76.83 | **75.00** | **91.59** | **71.35** | 83.99 | **74.35** |
| | MoCo | 800+100 | 70.64 | 41.65 | 87.64 | 39.31 | 92.12 | 75.03 | 73.94 | 89.53 | 68.31 | 83.55 | 72.34 |
| | MoCo + ArCL(views=2) | 800+100 | 70.26 | 41.86 | 89.52 | 40.21 | 92.64 | 75.73 | **74.04** | 88.97 | 70.06 | 84.31 | 73.04 |
| | MoCo + ArCL(views=3) | 800+100 | 70.92 | 43.87 | 89.36 | **42.37** | 93.30 | 76.93 | 72.93 | 90.50 | **71.14** | 84.03 | 73.83 |
| | MoCo + ArCL(views=4) | 800+100 | 69.42 | **45.55** | 89.61 | 41.91 | **93.55** | **77.05** | **74.04** | 91.17 | 70.93 | **85.48** | **74.37** |
| | MoCo | 200 | 67.72 | 40.02 | 86.59 | 37.41 | 90.90 | 72.43 | 73.88 | 87.97 | 66.97 | 80.00 | 70.69 |
| | MoCo + ArCL(views=2) | 200 | 68.04 | 42.34 | 87.92 | 36.45 | 92.29 | 71.71 | 74.68 | 89.00 | 67.87 | 81.42 | 71.85 |
| | MoCo + ArCL(views=3) | 200 | 68.74 | **43.21** | 88.26 | 38.27 | 92.49 | 75.02 | 74.68 | **89.61** | 68.31 | **81.64** | 72.39 |
| | MoCo + ArCL(views=4) | 200 | 68.92 | 41.65 | **88.42** | **38.77** | **92.70** | **75.73** | **75.43** | 88.95 | **68.63** | 81.60 | **72.43** |
| | **Supervised*** | | 77.20 | 43.59 | 90.18 | 44.92 | 91.42 | 73.90 | 72.23 | 89.93 | 69.49 | 91.45 | 74.12 |
| Finetune | MoCo | 800 | | 83.56 | 82.54 | 85.09 | 95.89 | 71.81 | 69.95 | 95.26 | 76.81 | 88.83 | 83.30 |
| | MoCo | 800+50 | | 83.15 | 84.50 | 85.90 | 96.13 | 72.58 | 70.16 | 94.44 | 79.34 | 86.12 | 83.59 |
| | MoCo + ArCL(views=2) | 800+50 | | **86.05** | 87.38 | **87.28** | 96.33 | 79.39 | 72.18 | 95.89 | 81.36 | 89.03 | 86.10 |
| | MoCo + ArCL(views=3) | 800+50 | | 84.03 | 87.64 | 86.34 | **96.88** | 80.98 | 72.87 | **96.14** | 81.90 | 89.20 | 86.22 |
| | MoCo + ArCL(views=4) | 800+50 | | 84.19 | **88.42** | 86.67 | 96.68 | **81.17** | **73.09** | 95.90 | 81.70 | **89.52** | **86.37** |
| | MoCo | 800+100 | | 83.18 | 84.50 | 84.27 | 96.01 | 72.14 | 70.27 | 95.53 | 78.23 | 88.73 | 83.65 |
| | MoCo + ArCL(views=2) | 800+100 | | 84.45 | 86.84 | 87.20 | 96.40 | 78.40 | 71.91 | 95.93 | 80.54 | 88.56 | 85.58 |
| | MoCo + ArCL(views=3) | 800+100 | | **85.94** | 86.85 | **87.34** | 96.36 | 79.75 | 71.44 | 96.00 | 81.48 | 88.26 | 85.94 |
| | MoCo + ArCL(views=4) | 800+100 | | 85.65 | **88.50** | 86.39 | **96.91** | 81.29 | **73.35** | 96.17 | 81.82 | **89.30** | **86.60** |
| | MoCo | 200 | | 83.18 | 82.66 | 84.47 | 95.51 | 72.54 | 70.43 | 94.99 | 77.39 | 86.12 | 83.03 |
| | MoCo + ArCL(views=2) | 200 | | 81.09 | 83.93 | **86.54** | 95.88 | 76.18 | **70.69** | 94.44 | 76.78 | 86.98 | 83.61 |
| | MoCo + ArCL(views=3) | 200 | | 84.79 | 85.61 | 85.39 | **96.56** | **78.81** | 70.59 | **95.84** | **80.71** | 87.91 | 85.13 |
| | MoCo + ArCL(views=4) | 200 | | **84.88** | **86.19** | 85.90 | 96.35 | 78.62 | **70.69** | 95.77 | 80.46 | **88.00** | **85.21** |
| | **Supervised*** | | | 83.50 | 91.01 | 82.61 | 96.39 | 82.91 | 73.30 | 95.50 | 84.60 | 92.42 | 86.92 |

fit on the extracted features. For full fine-tuning, the full models are trained for 5,000 steps using SGD with Nesterov momentum. We calculate the average results of the nine downstream datasets.

**Results.** We compare our methods with the MoCo baseline. There are about an average rise of 2% in the linear evaluation setting and about an average rise of 3% in the finetuning setting. In particular, we gain a huge improvement on CIFAR100 finetuning, where our proposed methods can outperform the baseline for up to 10%. It is also worthy to mention that our methods can reach the supervised results on nearly all the datasets. These facts can all indicate the effectiveness of our proposed methods.

Besides, the results also show that, under the same epoch setting, the average performance grows as the number of the views increase, which fits our theoretical analysis. One can also notice that there is only a small improvement in the $800 + 100$ epoch setting compared to the $800 + 50$ epoch setting. This implies that our methods can also converge in a rather fast speed, which can also be effective within a few epochs of training. Compared to CL methods, ArCL gets to see more augmentations in total since for each original sample, we construct its $m$ augmented versions.

### 6.3 COMPARISON WITH AVERAGE ALIGNMENT LOSS

Compared to CL methods, ArCL gets to see more augmentations in total since for each original sample, we construct its $m$ augmented versions. To make the comparison between ArCL and CL fairer, for original CL, we also construct $m$ views while use the average of the similarity between all positive pairs among the $m$ views, namely **average alignment loss** (AAL), as the learning objective. Results for SimCLR on CIFAR10 and CIFAR100 and MoCo on ImageNet can be seen in Table 5 and 6 in the appendix. Detailed experimental settings are also deferred there. As the experiments show, contrastive learning with AAL has similar performance with vanilla methods, which is still much worse than ArCL on distribution shift downstream tasks.

## ACKNOWLEDGEMENT

Yisen Wang is partially supported by the National Key R&D Program of China (2022ZD0160304), the National Natural Science Foundation of China (62006153), Open Research Projects of Zhejiang Lab (No. 2022RC0AB05), and Huawei Technologies Inc.

We would like to express our sincere gratitude to the reviewers of ICLR 2023 for their insightful and constructive feedback. Their valuable comments have greatly contributed to improving the quality of our work.

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

# Appendix

## A  MOCO + ARCL

---

**Algorithm 2:** MoCo + ArCL

---

**input** : Batch size $N$, temperature $\tau$, memory bank $M$, augmentation $\pi$, number of views $m$, epoch $T$, encoders $f_q, f_k$, projector $g$.

1 **for** $t = 1, \ldots, T$ **do**
2     sample minibatch $\{X_i\}_{i=1}^N$;
3     **for** $i = 1, \ldots, N$ **do**
4        draw $m$ augmentations $\widehat{\mathcal{A}} = \{A_1, \ldots, A_m\} \sim \pi$;
5        $z_{i,j}^1 = g(f_q(A_j X_i))$ for $j \in [m]$;
6        $z_{i,j}^2 = g(f_k(A_j X_i))$ for $j \in [m]$;
7        *# select the worst positive samples*;
8        $s_i^+ = \min_{j,l \in [m], j \neq l} \{z_{i,j}^{1\top} z_{i,l}^2 / (\|z_{i,j}^1\| \|z_{i,l}^2\|)\}$;
9        *# select the negative samples*;
10        **for** $m \in M$ **do**
11           $s_{i,m}^- = z_{i,1}^\top m / (\|z_{i,1}\| \|m\|)$;
12     compute $L = -\frac{1}{N} \sum_{i=1}^N \log \frac{\exp(s_i^+ / \tau)}{\sum_{m \in M} \exp(s_{i,m}^- / \tau)}$;
13     update $f$ and $g$ to minimize $L$;
14     update $M$;
15 **return** $f$

---

## B  PROOFS IN SECTION 3

*Proof of Lemma 3.1.* For simplicity we omit the notations $\mathcal{D}$ and $\vartheta$ in this proof, and use $x_1$ to denote the positive sample, i.e. the augmented data, use $p_k$ to denote $\mathcal{D}_\vartheta(C_k)$ and use $\mu_k$ to denote $\mu_k(f; \mathcal{D}_\vartheta)$. From Theorem 2 and Lemma B.1 in (Huang et al., 2021), we have

$$\mathbb{E}_{x \in C_k} \|f(x_1) - \mu_k\| \leq c\sqrt{\frac{1}{p_k}} L_{pos}^{\frac{1}{4}}(f) + \tau(\sigma, \delta) \tag{8}$$

for some constant $c$, where

$$\tau(\sigma, \delta) := 4\left(1 - \sigma\left(1 - \frac{L\delta}{4}\right)\right).$$

Note that $\tau$ is decreasing with $\sigma$ and increasing with $\delta$. Then we have

$$
\begin{aligned}
c\left(\sum_{k=1}^K \sqrt{p_k}\right) L_{\text{pos}}^{\frac{1}{4}}(f) + \tau(\sigma, \delta) &\geq \sum_{k=1}^K p_k \mathbb{E}_{x \in C_k} \|f(x_1) - \mu_k\| \\
&\geq \sum_{k=1}^K \frac{p_k}{\|h\|} \mathbb{E}_{x \in C_k} \mathbb{E}_{x_1 \in Ax} \|h \circ f(x_1) - h \circ \mu_k\| \\
&\geq \sum_{k=1}^K \frac{p_k}{\|h\|} \mathbb{E}_{x \in C_k} \mathbb{E}_{x_1 \in Ax} \|h \circ f(x_1) - e_k\| - \frac{1}{\|h\|} \sum_{k=1}^K p_k \|e_k - h \circ \mu_k\| \\
&= \frac{1}{\|h\|} R(h \circ f) - \frac{1}{\|h\|} \sum_{k=1}^K p_k \|e_k - h \circ \mu_k\|
\end{aligned}
\tag{9}
$$

for all linear layer $h \in \mathbb{R}^{K \times d_1}$. Therefore, we obtain

$$
\begin{aligned}
R(h \circ f) &\leq c\|h\| L_{\text{pos}}^{\frac{1}{4}} \sum_{k=1}^{K} \sqrt{p_k} + \|h\|\tau(\sigma,\delta) + \sum_{k=1}^{K} p_k \|e_k - h \circ \mu_k\| \\
&\leq c\|h\|\sqrt{K} L_{\text{pos}}^{\frac{1}{4}} + \|h\|\tau(\sigma,\delta) + \sum_{k=1}^{K} p_k \|e_k - h \circ \mu_k\|.
\end{aligned}
\tag{10}
$$

$\square$

*Proof of Theorem 3.2.* By the triangle inequality and equation 8, we have

$$
\|\mu_k\| \geq 1 - \mathop{\mathbb{E}}_{x \in C_k} \|f(x_1) - \mu_k\| \geq 1 - C\sqrt{\frac{1}{p_k}} L_{pos}^{\frac{1}{4}}(f) - \tau(\sigma,\delta).
$$

Therefore,

$$
\min_k \|\mu_k\|_2 \geq 1 - C\sqrt{\frac{1}{p_0}} L_{pos}^{\frac{1}{4}}(f) - \tau(\sigma,\delta).
$$

Define $U = (\mu_1, \ldots, \mu_K) \in \mathbb{R}^{d_1 \times K}$ and let $h_0 = U^+$ be the Moore–Penrose inverse of $U$ such that $h_0 U = I_K$. Then we have

$$
\|h_0\| = \sup_{x \in \mathbb{R}^{d_1}} \frac{\|x\|}{\|Ux\|}.
$$

For every $x = (x_1, \ldots, x_{d_1}) \in \mathbb{R}^{d_1}$ with $\|x\|_2 = 1$, we have

$$
\begin{aligned}
\|Ux\|_2^2 &= \sum_{i=1}^{d_1} x_i^2 \|\mu_i\|^2 + \sum_{i \neq j} x_i x_j \mu_i^\top \mu_j \\
&\geq \min_i \|\mu_i\|_2^2 - \max_{i,j} |\mu_i^\top \mu_j| \sum_{i \neq j} |x_i x_j| \\
&= \min_i \|\mu_i\|_2^2 - \max_{i,j} |\mu_i^\top \mu_j| \left( \left( \sum_i |x_i| \right)^2 - 1 \right) \\
&\geq \min_i \|\mu_i\|_2^2 - (K-1) \max_{i,j} |\mu_i^\top \mu_j| \\
&\geq 1 - c\sqrt{\frac{1}{p_0}} L_{\text{pos}}^{\frac{1}{4}}(f) - \tau(\sigma,\delta) - K \max_{i,j} |\mu_i^\top \mu_j| \\
&=: \frac{1}{\eta_{\text{reg}}(f;\mathcal{D},\vartheta)}.
\end{aligned}
$$

Therefore, we have

$$
\|h_0\| \leq \eta_{\text{reg}}(f;\mathcal{D},\vartheta).
$$

Then, by Lemma 3.1, we complete the proof by

$$
R(f) = \min_h R(h \circ f) \leq R(h_0 \circ f) \leq c\gamma_{reg}(f;\mathcal{D},\vartheta) L_{\text{pos}}^{\frac{1}{4}}(f;\mathcal{D},\vartheta) + \gamma_{reg}(f;\mathcal{D},\vartheta)\tau(\sigma,\delta).
$$

$\square$

## C  PROOF IN SECTION 4

*Proof of Proposition 4.1.* For any $\varepsilon > 0$, let $t = \sqrt{\varepsilon}/2$ and $f(x_1, x_2) = x_1 + tx_2$. Then, the alignment loss of $f$ satisfies

$$
\mathcal{L}_{\text{align}}(f;\mathcal{D},\pi) = t^2 \mathbb{E}\, X_2^2 \mathop{\mathbb{E}}_{(\theta_1,\theta_2) \sim \mathcal{N}(0,1)^2} (\theta_1 - \theta_2)^2 = 2t^2 < \varepsilon.
$$

Let $c = 0$ and $c' = 1/t$. Then obviously

$$
\mathcal{R}(f;\mathcal{D}_c) = 0,
$$

but

$$
\mathcal{R}(f;\mathcal{D}_{c'}) = P(X_1 < 0, X_1 + X_2 \geq 0) + P(X_1 \geq 0, X_1 + X_2 \leq 0) = \frac{1}{4}.
$$

$\square$

# D  PROOFS IN SECTION 5

*Proof of Theorem 5.1.* For any $f$ and $h$ with $\|h\| \le c_h$, we have

$$R(h \circ f; \mathcal{D}_A) - R(h \circ f; \mathcal{D}_{A'}) = \mathop{\mathbb{E}}_{(X,Y) \sim \mathcal{D}} \left( |h \circ f(AX) - Y|^2 - |h \circ f(A'X) - Y|^2 \right)$$

$$= \mathop{\mathbb{E}}_{(X,Y) \sim \mathcal{D}} (h \circ f(Ax) - h \circ f(A'x))((h \circ f(Ax) + h \circ f(A'x)) + 2y)$$

$$\le c \mathop{\mathbb{E}}_{(X,Y) \sim \mathcal{D}} \|h \circ f(Ax) - h \circ f(A'x)\|$$

$$\le c\|h\| \mathop{\mathbb{E}}_{(X,Y) \sim \mathcal{D}} \|f(Ax) - f(A'x)\|$$

$$\le c\|h\| L_{AR}(f)$$

for some constant $c$ which depends on $c_h$. $\square$

*proof of Theorem 5.2.* We first fix the sample $(x_1, \ldots, x_n)$, and consider the randomness of $\mathcal{A}_m(x_i)$. Define $A_1^*(x_i)$ and $A_2^*(x_i)$ as

$$A_1^*(x_i), A_2^*(x_i) = \arg \sup_{A_1, A_2 \in \mathcal{A}} \|f(A_1 x_i) - f(A_2 x_i)\|_2^2.$$

Note that their order doesn't matter. For every $\varepsilon > 0$, we have

$$P \left\{ \frac{1}{n} \sum_{i=1}^{n} \sup_{A_1, A_2 \in \mathcal{A}} \|f(A_1 x_i) - f(A_2 x_i)\|_2^2 \le \frac{1}{n} \sum_{i=1}^{n} \sup_{A_1, A_2 \in \mathcal{A}_m(x_i)} \|f(A_1 x_i) - f(A_2 x_i)\|_2^2 + L^2 L_{\mathcal{A}}^2 \left( \frac{\log(2n/\varepsilon)}{c_\vartheta m} \right)^{\frac{2}{d}} \right\}$$

$$\overset{(i)}{\ge} P \left\{ \max_{i \in [n]} \left\{ \inf_{A \in \mathcal{A}_m(x_i)} \|A - A_1^*(x_i)\|_2, \inf_{A \in \mathcal{A}_m(x_i)} \|A - A_2^*(x_i)\|_2 \right\} \le \left( \frac{\log(2n/\varepsilon)}{c_\vartheta m} \right)^{\frac{1}{d}} \right\}$$

$$\overset{(ii)}{\ge} 1 - \sum_{i=1}^{n} P \left\{ \inf_{A \in \mathcal{A}_m(x_i)} \|A - A_1^*(x_i)\|_2 \le \left( \frac{\log(2n/\varepsilon)}{c_\vartheta m} \right)^{\frac{1}{d}} \right\} - \sum_{i=1}^{n} P \left\{ \inf_{A \in \mathcal{A}_m(x_i)} \|A - A_2^*(x_i)\|_2 \le \left( \frac{\log(2n/\varepsilon)}{c_\vartheta m} \right)^{\frac{1}{d}} \right\}$$

$$\ge 1 - \sum_{i=1}^{n} P \left\{ \exists A \in \mathcal{A}_m(x_i): \|A - A_1^*(x_i)\|_2 \le \left( \frac{\log(2n/\varepsilon)}{c_\vartheta m} \right)^{\frac{1}{d}} \right\}$$

$$- \sum_{i=1}^{n} P \left\{ \exists A \in \mathcal{A}_m(x_i): \|A - A_2^*(x_i)\|_2 \le \left( \frac{\log(2n/\varepsilon)}{c_\vartheta m} \right)^{\frac{1}{d}} \right\}$$

$$\overset{(iii)}{\ge} 1 - 2n \left( 1 - \frac{c_\vartheta \log(2n/\varepsilon)}{c_\vartheta m} \right)^m$$

$$\overset{(iv)}{\ge} 1 - \varepsilon.$$

(i) comes from the Lipschitz continuity of $f$ and $A$; (ii) is a simple application of set operations; (iii) is derived from the fact that the volume of a $d$-dimensional ball of radius $r$ is proportional to $r^d$; (iii) comes from the inequality $(1 - 1/a)^a \le e^{-1}$ for $a > 1$. Therefore, with probability at least $1 - \varepsilon$, we have

$$\frac{1}{n} \sum_{i=1}^{n} \sup_{A_1, A_2 \in \mathcal{A}} \|f(A_1 x_i) - f(A_2 x_i)\|_2 \le \widehat{L}_{AR}(f) + L_f^2 L_{\mathcal{A}}^2 \left( \frac{\log(2n/\varepsilon)}{m} \right)^{\frac{2}{d}}.$$

Besides, for every $f$ we have

$$L_{AR}(f) \le \frac{1}{n} \sum_{i=1}^{n} \sup_{A_1, A_2 \in \mathcal{A}} \|f(A_1 x_i) - f(A_2 x_i)\|_2^2 + 2\mathfrak{R}_n(\mathcal{F}) + \left( \frac{\log(1/\varepsilon)}{n} \right)^{\frac{1}{2}}$$

with probability at least $1 - \varepsilon$. Then we finally obtain that with probability at least $1 - 2\varepsilon$, for every $f \in \mathcal{F}$,

$$L_{AR}(f) \le \widehat{L}_{AR}(f) + 2\widetilde{\mathfrak{R}}_n(\mathcal{F}) + \left( \frac{\log(1/\varepsilon)}{n} \right)^{\frac{1}{2}} + L_f^2 L_{\mathcal{A}}^2 \left( \frac{\log(2n/\varepsilon)}{c_\vartheta m} \right)^{\frac{2}{d}}.$$

$\square$

Table 4: Linear evaluation results of pretrained CIFAR10 models using MoCo v2 on CIFAR10, CIFAR100 and their modified versions.

|  | Method | Aug 1 | Aug 2 | Aug 3 | Aug 4 | Aug 5 | Original |
|---|---|---|---|---|---|---|---|
| CIFAR10 | MoCo | 88.34 | 87.44 | 89.14 | 88.76 | 88.12 | 89.29 |
|  | MoCo + ArCL (views=2) | 89.12 | 88.13 | 89.62 | 89.25 | 88.91 | 90.44 |
|  | MoCo + ArCL (views=3) | 90.22 | 89.11 | 90.93 | 90.13 | 89.75 | 91.22 |
|  | MoCo + ArCL (views=4) | **90.77** | **89.64** | **91.22** | **90.86** | **90.18** | **91.45** |
| CIFAR100 | SimCLR | 52.98 | 48.77 | 54.47 | 53.18 | 52.93 | 55.88 |
|  | SimCLR+ArCL(views=2) | 53.37 | 49.14 | 55.09 | 53.46 | 53.14 | 56.62 |
|  | SimCLR+ArCL(views=3) | 54.49 | 50.29 | 55.33 | **53.78** | 53.62 | 57.13 |
|  | SimCLR+ArCL(views=4) | **55.42** | **51.42** | **55.96** | 53.76 | **54.41** | **57.98** |

## E  ADDITIONAL EXPERIMENTS

In order to further verify the generality of our method, we conduct more experiments on different settings and compare our objective with different loss.

### E.1  MOCO ON CIFAR10

**Setup.** In this part, we conduct experiments with MoCo v2 pretrained on CIFAR10. We follow the setup of MoCo v2. We use ResNet-18 as the encoder and train the representation with 400 epochs. The temperature is set to 0.2 and the initial learning rate is set to 0.15. The memory bank size is set to 4096. The SGD optimizer and a cosine schedule for learning rate are used. Warm-up is not used. We conduct linear evaluation with 100 epochs on modified CIFAR10 and CIFAR100 which are used in Section 6.1.

**Results.** Results can be seen in Table 4. We can see that the original results still maintain in the MoCo method. Our proposed approach improves the transferability of MoCo. As the number of views grows, the accuracy also increases.

### E.2  COMPARISON WITH AVERAGE ALIGNMENT LOSS

To make the comparison between ArCL and CL fairer, we add new experiments for original CL. For each sample, we also construct $m$ views and use the expectation of the similarity between positive pairs, namely **average alignment loss**(AAL), as the learning objective. The settings and results are shown in the following part.

**Loss Function.** For image $x_i$, the normalized features of its $m$ views from the online branch are $z_{i1}, z_{i2}, \ldots, z_{im}$ and the normalized features from the target branch are $z'_{i1}, z'_{i2}, \ldots, z'_{im}$. Our ArCL alignment loss is

$$L_{ArCL}^{align} = -\sum_i \min_{j \neq k} z_{ij}^\top z'_{ik}/\tau.$$

And the average alignment loss should be

$$L_{Average}^{align} = -\sum_i \underset{j \neq k}{\mathrm{avg}}\, z_{ij}^\top z'_{ik}/\tau = -\sum_i (\sum_{j \neq k} z_{ij}^\top z'_{ik}/\tau)/(m^2 - m).$$

The uniformity loss keeps the same and the total loss is the sum of the alignment loss and the uniformity loss.

**Hyperparameters and Datasets.** For SimCLR, we train the models on CIFAR10 and evaluate them on modified CIFAR10 and CIFAR100 just as our original paper does. We choose the augmentation views to be 4 and the batch size to be 512. For MoCo, We conduct experiments on ImageNet, with the 800-epochs-pretrained model as the initializing model. We train the model for 50 epochs and use the same setting as in Section 6.2.

**Results.** Results for SimCLR experiments can be seen in Table 5. We compare the linear evaluation results on modified CIFAR10 and CIFAR100. Results for MoCo experiments can be seen in 6. We compare both the linear evaluation and finetuning results on small datasets. As the experiments

Table 5: Linear evaluation results of pretrained CIFAR10 models using SimCLR with two different alignment losses on CIFAR10, CIFAR100 and their modified versions.

| | Method | Aug 1 | Aug 2 | Aug 3 | Aug 4 | Aug 5 | Original |
|---|---|---|---|---|---|---|---|
| CIFAR10 | SimCLR | 88.62 | 86.27 | 88.96 | 88.56 | 88.37 | 88.81 |
| | SimCLR + AAL (views=4) | 88.90 | 86.34 | 88.72 | 88.65 | 88.44 | 89.97 |
| | SimCLR + ArCL (views=4) | **89.97** | **88.06** | **90.48** | **89.91** | **89.59** | **90.20** |
| CIFAR100 | SimCLR | 52.28 | 48.09 | 53.45 | 52.58 | 51.53 | 53.12 |
| | SimCLR+AAL(views=4) | 52.34 | 48.77 | 53.12 | 52.44 | 51.82 | 53.21 |
| | SimCLR+ArCL(views=4) | **53.40** | **50.16** | **54.92** | **53.77** | **52.61** | **54.20** |

Table 6: Results comparison on linear evaluation (up) and finetuning (down) of pretrained ImageNet models with ArCL alignment loss and average alignment loss on popular recognition datasets. Results style: **best** in the same augmentation view setting. **Avg** takes the average of the results on all the nine small datasets.

| | Epochs | ImageNet | Aircraft | Caltech101 | Cars | CIFAR10 | CIFAR100 | DTD | Flowers | Food | Pets | **Avg** |
|---|---|---|---|---|---|---|---|---|---|---|---|---|
| Linear | MoCo | 70.68 | 41.79 | 87.92 | 39.31 | 92.28 | 74.90 | 73.88 | 90.07 | 68.95 | 83.30 | 72.49 |
| | MoCo + AAL(views=2) | **71.12** | 40.53 | 87.80 | 38.64 | 92.23 | 75.14 | **74.95** | 88.64 | 69.24 | 83.17 | 72.26 |
| | MoCo + ArCL(views=2) | 69.70 | **44.29** | **89.79** | **42.15** | **93.07** | **76.70** | 74.20 | **90.40** | **70.94** | **83.68** | **73.91** |
| | MoCo + AAL(views=3) | **71.37** | 40.41 | 87.79 | 42.09 | 92.64 | 75.31 | 74.89 | 89.23 | 69.37 | 83.79 | 72.84 |
| | MoCo + ArCL(views=3) | 69.80 | **44.57** | **89.48** | **42.11** | **93.29** | **77.33** | **74.63** | **91.13** | **71.16** | **84.23** | **74.21** |
| Finetune | MoCo | | 83.56 | 82.54 | 85.09 | 95.89 | 71.81 | 69.95 | 95.26 | 76.81 | 88.83 | 83.30 |
| | MoCo + AAL(views=2) | | 83.87 | 82.76 | 85.90 | **96.38** | 71.43 | **72.71** | 95.50 | 76.95 | **89.05** | 83.84 |
| | MoCo + ArCL(views=2) | | **86.05** | **87.38** | **87.28** | 96.33 | **79.39** | 72.18 | **95.89** | **81.36** | 89.03 | **86.10** |
| | MoCo + AAL(views=3) | | 83.07 | 83.21 | 85.19 | 96.37 | 72.02 | 72.55 | 95.74 | 79.62 | 88.83 | 84.07 |
| | MoCo + ArCL(views=3) | | **84.03** | **87.64** | **86.34** | **96.88** | **80.98** | **72.87** | **96.14** | **81.90** | **89.20** | **86.22** |

show, when contrastive learning methods with average alignment loss use the same amount of data as ArCL, although they can have higher accuracy compared to vanilla methods (those which only use two views) on ID set, they still have a similar performance with vanilla methods on OOD set, which is much worse than ArCL. This further verifies the superiority of our method.

### E.3   EXPERIMENTS ON MNIST-CIFAR

MNIST-CIFAR is a synthetic dataset proposed by (Shah et al.). It consists of $3 \times 64 \times 32$ synthetic images, each of which is a vertical concatenation of a $3 \times 32 \times 32$ MNIST image from class 0 or class 1, and a $3 \times 32 \times 32$ CIFAR10 image from class 0 or class 1. We follow the distribution settings proposed by (Shi et al., 2022):

- ID train: 1.0 correlation between MNIST and CIFAR10 labels. Contains two classes: class 0 with MNIST "0" and CIFAR10 "automobile", and class 1 with MNIST "1" and CIFAR10 "plane".

- OOD train: no correlation between MNIST and CIFAR10 labels, images from the two classes of MNIST and CIFAR10 are randomly paired. We choose the label of CIFAR10 to be the label of the concatenated image.

- OOD test: generated similarly to the OOD train set using the test set of MNIST and CIFAR10.

In this experiment, we train the model on the ID train set, conduct the linear evaluation on the OOD train set, and calculate the accuracy on the OOD test set. We adopt SimCLR as the CL framework and use the 4-layer CNN as the backbone just as Shi et al. (2022) does. We fix the base feature size of CNN $C = 32$ and the latent dimension $L = 128$. Batch size is set to 128, the optimizer is set to SGD and the lr. scheduler is set to warmup cosine. The learning rate and weight decay are searched in the range given in Table 2 of Shi et al. (2022). Since the training process is easy to converge, we set the training epoch and the linear evaluation epoch to 5. We also compare the average alignment loss (AAL for short) mentioned in the Appendix 6.3.

Table 7: Linear evaluation results of pretrained models using SimCLR with two different alignment losses on MNIST-CIFAR dataset. The average results under three diffrent random seeds are given.

| Methods | Accuracy(%) | Methods | Accuracy(%) |
|---|---|---|---|
| SimCLR | 85.6 | | |
| SimCLR + ArCL(views=3) | 86.0 | SimCLR+AAL(views=3) | 85.4 |
| SimCLR + ArCL(views=4) | 87.2 | SimCLR+AAL(views=4) | 86.1 |
| SimCLR + ArCL(views=5) | 87.3 | SimCLR+AAL(views=5) | 86.3 |
| SimCLR + ArCL(views=6) | 88.4 | SimCLR+AAL(views=6) | 85.8 |

**Results.** We can see that by using ArCL, SimCLR enjoys a rising performance. The accuracy also grows as the number of views grows, which fits our theory. We also notice that the accuracy of SimCLR with AAL does not vary too much as the number of views grows, which indicates that the key point is the usage of multi-view (**adopting the minimum** instead of the average).

