# OpenReview forum: "ArCL: Enhancing Contrastive Learning with Augmentation-Robust Representations"
_ICLR.cc/2023/Conference — ICLR 2023 poster_

### Official Review · Reviewer_jXHS · 2022-10-23

**Confidence:** 4
**Correctness:** 3
**Technical Novelty And Significance:** 2
**Empirical Novelty And Significance:** 3
**Recommendation:** 5

**Clarity, Quality, Novelty And Reproducibility:**

* **Clarity** - The theoretical results are a bit opaque. Its implications need to be clarified.

* **Reproducibility** - I do not have any concerns regarding reproducibility

* ** Novelty** - The proposed regulariser ArCL is novel, as far as I am aware. However, the drawn conclusion about more augmentations helping SSL is not novel.

**Strength And Weaknesses:**

# Strength
* The paper studies the role of data augmentations in contrastive SSL in a rigorous manner. I wonder if it can be linked to the concept of expansion [3]. This can serve as a basis for more theoretical studies on the role of data-augmentations in various fields within ML.

* The concept of ArCL is quite natural but showing how it might be necessary is an interesting take.

# Cons

* As I explain below, the experimental comparisons between ArCL and SImCLR might be unfair. Further, if the topic is distribution shift, I would recommend using more distribution shifts for example _Controllable Shift_s in [1].
* The significance of the theoretical results are not immediately clear. Ofcourse, the accuracy increases with more augmentations , but other than that there does not seem to be any takeaway from the theoretical results. Also some terms, like $n$  and the absence of $N$ in Thm 5.2 is unclear.
* Discussion of similar work like [2] is also missing. This particular work also discusses the same question but highlights the importance of the number of negative samples, which is completely ignored in this work.

**Summary Of The Paper:**

The paper studies the topic of contrastive SSL. In particular, the paper studies whether using multiple data augmentations and choosing the one the model performs worst at is a more effective approach than Contrastive SSL. The paper calls this ArCL and also provides more experiments.

**Summary Of The Review:**

* In Equation (2), the linear classifier is evaluated as the best linear classifier on the target domain. While this is usually not done in practice, some recent works (e.g. [1]) used this to empirically observe the robustness of SSL methods. The authors should discuss this and other similar works as they provide empirical  use cases of their setting. Perhaps, Lemma 3.1 indeed applies to the experimental setting in these papers.

* Theorem 3.2 shows a kind of equivalence between SSL and Supervised for appropriately chosen regularisation term, This is similar to the recent work of bao et. al, which is also not cited in this paper. In addition Bao et. al. quantifies the effect of the number of negative samples in the regularisation term and presents a more realistic results as increasing number of negative examples work better in practice.

* The example in Proposition 4.1 presents a multiplicative noise model. This can indeed lead to interesting theoretical examples showing that minimising expectation across domains is arbitrarily worse than  minimising supremum across domains. However, I am not sure if this is a realistic problem to address.

* Theorem 5.2 has a dependence on m and n, but I don't see the dependence on N, the number of negative examples. I am also not sure what $n$ is here. It seems to be an application of Rademacher uniform convergence result, but $n$ is not defined here. Could the author discuss what this is ?

* In the experiments, doing an ArCL with m views requires m forward passes with m augmentations and then selecting the worst one. A normal SimCLR uses just one random augmentation instead if I understand correctly. That is an unfair comparison. Could the authors do a fairer comparison either by multiplying the number of epochs with the number of views or doing an expectation over m views for SimCLR.

* Could the authors also more extreme distribution shift experiments on more realistic/synthetic distribution shifts using the framework provided in [1] ?

* How do the results change with varying $k$ as in [2] ?


=========== Post Rebuttal =================

As the discussion below shows, we had an interesting discussion with the authors and due to the reasons stated below, I am of the position that I keep the score but do object acceptance if the other reviewers want to fight for it.


[1] How robust are pre-trained models to distribution shift? https://arxiv.org/abs/2206.08871
[2] Bao, Han, Yoshihiro Nagano, and Kento Nozawa. "Sharp learning bounds for contrastive unsupervised representation learning." arXiv preprint arXiv:2110.02501 (2021).
[3] Balcan, Maria-Florina, Avrim Blum, and Ke Yang. "Co-training and expansion: Towards bridging theory and practice." Advances in neural information processing systems 17 (2004).

---

> ### Author Response · Authors · 2022-11-16
> **Response to Reviewer jXHS (4/4)**
>
>
>
> **Q5.** In Equation (2), the linear classifier is evaluated as the best linear classifier on the target domain. The authors should discuss this and other similar works as they provide empirical use cases of their setting.
>
> **A5.** We thank you for providing a related work [1] that we omitted. As you point out, this paper provides empirical use cases of their setting. We have cited and discussed it in our revised paper.
>
> References:
>
> [1] Shi, Yuge, et al. "How robust are pre-trained models to distribution shift?." arXiv preprint arXiv:2206.08871 (2022).
>
> ---
>
> **Q7.** Could the authors also more extreme distribution shift experiments on more realistic/synthetic distribution shifts using the framework provided in [1]?
>
> **A7.** We notice that the MNIST-CIFAR setting in [1] is suitable to apply. The MNIST-CIFAR dataset consists of $3\times 64\times 32$ synthetic images, each of which is a vertical concatenation of a $3\times32\times32$ MNIST image from class 0 or class 1, and a $3\times 32\times 32$ CIFAR10 image from class 0 or class 1. The distribution settings are illustrated below:
>
> - **ID train:** 1.0 correlation between MNIST and CIFAR10 labels. Contains two classes: class 0 with MNIST “0” and CIFAR10 “automobile”, and class 1 with MNIST “1” and CIFAR10 “plane”.
> - **OOD train:** no correlation between MNIST and CIFAR10 labels, images from the two classes of MNIST and CIFAR10 are randomly paired. We choose the label of CIFAR10 to be the label of the concatenated image.
> - **OOD test:** generated similarly to the OOD train set using the test set of MNIST and CIFAR10.
>
> In this experiment, we train the model on the ID train set, conduct the linear evaluation on the OOD train set, and calculate the accuracy on the OOD test set. We adopt SimCLR as the CL framework and use the 4-layer CNN as the backbone just as [1] does. We fix the base feature size of CNN $C=32$ and the latent dimension $L=128$. Batch size is set to 128, the optimizer is set to SGD and the lr. scheduler is set to warmup cosine. The learning rate and weight decay are searched in the range given in Table 2 of [1]. Since the training process is easy to converge, we set the training epoch and the linear evaluation epoch to 5. We also compare the average alignment loss (AAL for short) mentioned in the above **Answer 2**. The average results on the OOD test set under three different seeds are shown below.
> |Methods|Accuracy(\%)|Methods|Accuracy(\%)
> |---|---|---|---|
> |SimCLR|85.6|
> |SimCLR+ArCL(views=3)|86.0|SimCLR+AAL(views=3)|85.4|
> |SimCLR+ArCL(views=4)|87.2|SimCLR+AAL(views=4)|86.1|
> |SimCLR+ArCL(views=5)|87.3|SimCLR+AAL(views=5)|86.3|
> |SimCLR+ArCL(views=6)|88.4|SimCLR+AAL(views=6)|85.8|
>
>
> We can see that by using ArCL, SimCLR enjoys a rising performance. The accuracy also grows as the number of views grows, which fits our theory. We also notice that the accuracy of SimCLR with AAL does not vary too much as the number of views grows, which indicates that the key point is the usage of multi-view (**adopting the minimum** instead of the average). These experiments are also shown in Appendix E.
>
>
> [1] Shi, Yuge, et al. "How robust are pre-trained models to distribution shift?." arXiv preprint arXiv:2206.08871 (2022).
>
> ---
>
> **Q8.** How do the results change with varying $k$ as in [2] ?
>
> **A8.** By studying the gap between contrastive loss and supervised loss, [2] concluded that a larger negative sample size $N$ is beneficial for contrastive learning. Although our paper does not focus on the role of negative samples, **our result coincides with [2]**. That can be seen from the term $\gamma_{\mathrm{reg}}$ appearing in Thm 3.2. In principle, the divergence term $-\max_{k \neq k^{\prime}}\left|\mu_k\left(f;\mathcal{D}\_\pi \right)^\top \mu_{k^\prime}\left(f ; \mathcal{D}\_\pi\right)\right|$ in $\gamma_{\mathrm{reg}}$ increases when $N$, and hence $\gamma_{\mathrm{reg}}^{-1}$ decreases. This leads to better risk bound in Thm 3.2. A rigorous result that involves $N$ explicitly needs additional effort, and is not the focus of our work.
>
> References:
>
> [2] Bao, Han, Yoshihiro Nagano, and Kento Nozawa. "Sharp learning bounds for contrastive unsupervised representation learning." arXiv preprint arXiv:2110.02501 (2021).
>
> ---
>
> Hope our detailed explanations and experiments above could help address your concerns. We are looking forward to your reply and please let us know if there is more to clarify.

---

> > ### Comment · Reviewer_jXHS · 2022-11-27
> > **Reply**
> >
> > I thank the author's for the additional experiments and efforts put in the rebuttal. While some of my questions have been clarified e g. with additional discussions some still remain.
> >
> > 1. I had mentioned the controllable shift datasets in my review which are more realistic and not the MNIST-CIFAR that the authors have conducted.
> >
> > 2. I still cannot quite understand where the number of negative samples come in. We know, from existing theory and practice, that they play an important role. It's omission from the results here makes it confusing for me to interpret
> >
> > 3. Under fairer comparison with SimCLR and Moco, the advantages of using ArCL seems to diminish to a very small (almost statistically insignificant) amount.
> >
> > Thus, I'm unwilling to change my score but if the other reviewers push for acceptance despite these comments, I won't fight back.

---

> > > ### Author Response · Authors · 2022-12-09
> > > **Response to Reviewer jXHS**
> > >
> > > Thanks for your further comments.
> > >
> > > * We will conduct new experiments on more realistic datasets. Nevertheless, our new experiments on MNIST-CIFAR at least show that our ArCL can still work well under this extreme domain shift.
> > > * Although our paper does not focus on the role of negative samples, our result coincides with other works, i.e, more negative samples is better. That can be seen from the term appearing in Thm 3.2. In principle, $\gamma_{\mathrm{reg}}$ increases with the negative sample size, and hence the risk bound in Thm 3.2 becomes better.
> > > * For the fairer comparison with SimCLR and MoCo, we want to argue that the **improvement of ArCL is still significant**. In linear evaluation, the average accuracy of MoCo+3 views is **72.84**, while the ArCL+3 views is **74.21**. In full fine-tuning, the average accuracy of MoCo+3 views is **84.07**, while the ArCL+3 views is **86.22**. We think this improvement is significant.
> > >
> > > Thanks again for your time and detailed comments!

---

> ### Author Response · Authors · 2022-11-16
> **Response to Reviewer jXHS (3/4)**
>
> **Q3.** The significance of the theoretical results is not immediately clear. Of course, the accuracy increases with more augmentations, but other than that there does not seem to be any takeaway from the theoretical results. Theorem 5.2 has a dependence on m and n, but I don't see the dependence on N, the number of negative examples. I am also not sure what n is here.
>
> **A3.** The term **$n$ denotes the number of unlabeled training data**, which is defined before equation (7). To avoid confusion, we have added its meaning in Theorem 5.2 in the revised version. We remark that Theorem 5.2 is **not about the surrogate gap** between the supervised and contrastive loss. It is to **bound the gap between $\mathcal{L}\_{\mathrm{AR}}$ and $\widehat{\mathcal{L}}\_{\mathrm{AR}}$**, where the latter is the empirical augmentation-robust loss used actually in ArCL, and the former is its expectation. Since this loss **involves only positive samples**, the number of negative samples $N$ does appear in it. The conclusion of Theorem 5.2 is that when $m$ increases, the **approximation to the supremum** in $\mathcal{L}\_{\mathrm{AR}}$ is better, and is not about the performance of CL studied in [2] and many other works.
>
> ---
>
> **Q4.** Discussion of similar work like [2] is also missing. This particular work also discusses the same question but highlights the importance of the number of negative samples.
>
> **Q6.** Theorem 3.2 shows a kind of equivalence between SSL and Supervised for appropriately chosen regularisation terms, This is similar to the recent work of bao et. al, which is also not cited in this paper.
>
> **A4&6.** We thank you for providing the related work that we omitted. **We have cited and discussed it in our revised paper.** We give a more detailed discussion here.
>
> [2] established surrogate upper and lower bounds for the downstream classification loss, which applies to any negative sample size $N$ and best explains the empirical observations in the earlier studies. Although both our [2] and our paper derive some downstream classification loss bounds, we highlight some significant differences in the following.
> * **Different data generation model.** [2] assumed that positive pairs are drawn from the same latent classes, not generated by data augmentation. Although this assumption was adopted by many contrastive learning theories, it is too ideal since it essentially requires label information. To study the role of data augmentation, **instead of imposing this assumption**, we use the **$(\sigma,\delta)$-concentration** to quantify the goodness of data augmentation, and show how it affects transferability.
>
> References:
>
> [2] Bao, Han, Yoshihiro Nagano, and Kento Nozawa. "Sharp learning bounds for contrastive unsupervised representation learning." arXiv preprint arXiv:2110.02501 (2021).

---

> ### Author Response · Authors · 2022-11-16
> **Response to Reviewer jXHS (2/4)**
>
> **Q2.** In the experiments, doing an ArCL with $m$ views requires $m$ forward passes with $m$ augmentations and then selecting the worst one. Could the authors do a fairer comparison either by multiplying the number of epochs with the number of views or doing an expectation over $m$ views for SimCLR?
>
> **A2.** To make the comparison between ArCL and CL fairer, we add new experiments for CL. For each sample, we also construct $m$ views and use the expectation of the similarity between positive pairs, namely **average alignment loss**(AAL), as the learning objective. The settings and results are shown in the following.
>
> **Experiment Settings:**
> - **Loss Function:** For image $x_i$, the normalized features of its $m$ views from the online branch are $z_{i1},z_{i2},\dots,z_{im}$ and the normalized features from the target branch are $z'\_{i1},z'\_{i2},\dots,z'\_{im}$. Our ArCL alignment loss is
> $$L^{align}\_{ArCL}=-\sum\_i\min\_{j\neq k}z^\top\_{ij}z\_{ik}'/\tau.$$
> And the average alignment loss should be
> $$L^{align}\_{Average}=-\sum\_i \mathop{\text{avg}}\_{j\neq k}\ z^\top\_{ij}z\_{ik}'/\tau=-\sum\_i(\sum\_{j\neq k}z^\top\_{ij}z\_{ik}'/\tau)/(m^2-m).$$
> The uniformity loss keeps the same and the total loss is the sum of the alignment loss and the uniformity loss.
> - **Hyperparameters and Datasets:** **For SimCLR**, we train the models on CIFAR10 and evaluate them on modified CIFAR10 and CIFAR100 just as our original paper does. We choose the augmentation views to be 4 and the batch size to be 512. **For MoCo**, We conduct the experiment on ImageNet, with the 800-epochs-pretrained model as the initializing model. We train the model for 50 epochs and use the same setting as mentioned in our paper.
> - **Results for SimCLR:** We compare the linear evaluation results on modified CIFAR10 and CIFAR100.
>
> ||Method|Aug 1|Aug 2|Aug 3|Aug 4|Aug 5|Original|
> |---|---|---|---|---|---|---|---|
> |CIFAR10|SimCLR|88.62|86.27|88.96|88.56|88.37|88.81|
> ||SimCLR+Average Alignment Loss(views=4)|88.90|86.34|88.72|88.65|88.44|89.97
> ||SimCLR+ArCL(views=4)|**89.97**|**88.06**|**90.48**|**89.91**|**89.59**|**90.20**
> |CIFAR100|SimCLR|52.28|48.09|53.45|52.58|51.53|53.12|
> ||SimCLR+Average Alignment Loss(views=4)|52.34|48.77|53.12|52.44|51.82|53.21|
> ||SimCLR+ArCL(views=4)|**53.40**|**50.16**|**54.92**|**53.77**|**52.61**|**54.20**
>
>
>
>
> - **Results for MoCo:** We compare the linear evaluation results and finetuning results.
>
> |||ImageNet|Aircraft|Caltech101|Cars|CIFAR10|CIFAR100|DTD|Flowers|Food|Pets|**Avg**
> |:---:|:---------:|:-----:|:-----:|:-----:|:-----:|:------:|:------:|:-----------------:|:--------:|:------:|:------:|:-----:|
> |Linear|MoCo|70.68|41.79|87.92|39.31|92.28|74.90|73.88|90.07|68.95|83.30|72.49|
> ||MoCo+Average Alignment Loss(views=2)|**71.12**|40.53|87.80|38.64|92.23|75.14|**74.95**|88.64|69.24|83.17|72.26
> ||MoCo+ArCL(views=2)|69.70|**44.29**|**89.79**|**42.15**|**93.07**|**76.70**|74.20|**90.40**|**70.94**|**83.68**|**73.91**
> ||MoCo+Average Alignment Loss(views=3)|**71.37**|40.41|87.79|42.09|92.64|75.31|74.89|89.23|69.37|83.79|72.84
> ||MoCo+ArCL(views=3)|69.80|**44.57**|**89.48**|**42.11**|**93.29**|**77.33**|**74.63**|**91.13**|**71.16**|**84.23**|**74.21**
> |Finetune|MoCo||83.56|82.54|85.09|95.89|71.81|69.95|95.26|76.81|88.83|83.30
> ||MoCo+Average Alignment Loss(views=2)||83.87|82.76|85.90|**96.38**|71.43|**72.71**|95.50|76.95|**89.05**|83.84
> ||MoCo+ArCL(views=2)||**86.05**|**87.38**|**87.28**|96.33|**79.39**|72.18|**95.89**|**81.36**|89.03|**86.10**|
> ||MoCo+Average Alignment Loss(views=3)||83.07|83.21|85.19|96.37|72.02|72.55|95.74|79.62|88.83|84.07|
> ||MoCo+ArCL(views=3)||**84.03**|**87.64**|**86.34**|**96.88**|**80.98**|**72.87**|**96.14**|**81.90**|**89.20**|**86.22**
>
> - As the experiments show, when contrastive learning methods with average alignment loss use the same amount of data as ArCL, although they can have higher accuracy compared to vanilla methods (those which only use two views) on ID set, they still have a similar performance with vanilla methods on OOD set, which is much worse than ArCL. This further verifies the superiority of our method. These experiments are also shown in Appendix E.
>
> ---

---

> > ### Comment · Reviewer_jXHS · 2022-11-27
> > **Results do not seem to justify ArCL's claimed superiority u der fairer comparison**
> >
> > Unless I'm missing something, the above experiments with the fairer loss function for contrastive method does not show much of a difference between ArCL and existing methods.

---

> ### Author Response · Authors · 2022-11-16
> **Response to Reviewer jXHS (1/4)**
>
> We thank Reviewer jXHS for careful reading and constructive comments, though there might be some misunderstandings of our contributions.  We will address your concerns in the following points.
>
> ---
>
> **Q1.** I wonder if the study can be linked to the concept of expansion [3]. This can serve as a basis for more theoretical studies on the role of data augmentations in various fields within ML.
>
> **A1.** [3] studied co-training, which is a classical version of self-training that requires two distinct views. They introduce the "expansion" to characterize the distribution of two-view data. If we regard the positive pairs in self-supervised learning as two different views, then **the two settings are similar.** However, although both the "expansion" and our $(\sigma,\delta)$-notion are used to model the two-view data, **they are local and global measures respectively, and thus fundamentally different**. Nevertheless, as you point out, "expansion" can indeed be used to study the role of data augmentation in a variety of domains. For example, [4] used "expansion" to study self-training and unsupervised learning. For more details, please refer to it.
>
> References:
>
> [3] Balcan, Maria-Florina, Avrim Blum, and Ke Yang. "Co-training and expansion: Towards bridging theory and practice." Advances in neural information processing systems 17 (2004).
> [4] Wei, Colin, et al. "Theoretical analysis of self-training with deep networks on unlabeled data." arXiv preprint arXiv:2010.03622 (2020).
>
> ---

---

### Official Review · Reviewer_y4QB · 2022-10-24

**Confidence:** 2
**Correctness:** 4
**Technical Novelty And Significance:** 3
**Empirical Novelty And Significance:** 2
**Recommendation:** 6

**Clarity, Quality, Novelty And Reproducibility:**

The paper is easy to follow and focuses on theoretical analysis of contrastive loss in self-supervised learning, followed with an empirical improvement.

**Strength And Weaknesses:**

+ This work starts from a theoretical perspective to analyze the OOD performance of contrastive learning in self-supervised learning. The analysis reveals that the better performance of contrastive learning mainly comes from data augmentation.
+ This work proposes a simple augmentation-robust contrastive learning to improve OOD generalization of SSL.
- The empirical results seem not to be very extensive. Can authors provide the results of ArCL applied to both SimCLR and MoCo on all the three datasets rather than picking one model for each of these datasets?

**Summary Of The Paper:**

This work established a theoretical framework to analyze the OOD performance of contrastive learning based self-supervised learning and suggest that the effective of contrastive learning mainly comes from data augmentation. Further, they proposed augmentation-robust contrastive learning to show the better OOD performance.

**Summary Of The Review:**

Since I do not have too much background in theoretical analysis, I could hardly measure the novelty and contribution of this work to the community. The proposed augmentation-robust contrastive learning (ArCL) seems to work well on two framework SimCLR and Moco although I am not sure why the authors do not provide the two frameworks on all the three datasets.

---

> ### Author Response · Authors · 2022-11-16
> **Response to Reviewer y4QB**
>
> We thank Reviewer y4QB for appreciating our work. We will address your main concerns in the following point.
>
> ---
>
> **Q1.** The empirical results seem not to be very extensive. Can authors provide the results of ArCL applied to both SimCLR and MoCo on all three datasets rather than picking one model for each of these datasets?
>
> **A1.** Thanks for this suggestion. We have conducted experiments for MoCo + ArCL on CIFAR10 and CIFAR100 to further verify the generality of our method.
>
> - **Experiment Settings**: We follow the setup of MoCo v2. We use ResNet-18 as the encoder and train the representation with 400 epochs. The temperature is set to 0.2 and the initial learning rate is set to 0.15. The memory bank size is set to 4096. The SGD optimizer and a cosine schedule for learning rate are used. Warm-up is not used. We conduct linear evaluation with 100 epochs on modified CIFAR10 and CIFAR100.
>
> - **Results:**
>
> ||Method|Aug 1|Aug 2|Aug 3|Aug 4|Aug 5|Original|
> |---|---|---|---|---|---|---|---|
> |CIFAR10|MoCo|88.34|87.44|89.14|88.76|88.12|89.29|
> ||MoCo+ArCL(views=2)|89.12|88.13|89.62|89.25|88.91|90.44
> ||MoCo+ArCL(views=3)|90.22|89.11|90.93|90.13|89.75|91.22
> ||MoCo+ArCL(views=4)|**90.77**|**89.64**|**91.22**|**90.86**|**90.18**|**91.45**
> |CIFAR100|MoCo|52.98|48.77|54.47|53.18|52.93|55.88|
> ||MoCo+ArCL(views=2)|53.37|49.14|55.09|53.46|53.14|56.62|
> ||MoCo+ArCL(views=3)|54.49|50.29|55.33|**53.78**|53.62|57.13|
> ||MoCo+ArCL(views=4)|**55.42**|**51.42**|**55.96**|53.76|**54.41**|**57.98**
>
> We can see that the original results still maintain in the MoCo method. Our proposed approach improves the transferability of MoCo. As the number of views grows, the accuracy also increases.
>
> SimCLR on ImageNet needs a large batch size to perform, which is computationally hard and time-consuming. We will verify our method in this setting as soon as we can in the future. Please refer to Appendix E for more extensive experiments on different settings.
>
> ---
>
> Thanks again for your comments and hope our answers could address your concerns. Please let us know if there is more to clarify.

---

### Official Review · Reviewer_QCLS · 2022-10-26

**Confidence:** 4
**Correctness:** 2
**Technical Novelty And Significance:** 3
**Empirical Novelty And Significance:** 2
**Recommendation:** 6

**Clarity, Quality, Novelty And Reproducibility:**

The paper is mostly easy to follow and many of the findings, especially the use of worst case augmentation unalignment, are novel. Some of the claims are not well justified, as described above. Also the use of the term *OOD generalization* seems incorrect for the setting that is considered.

**Strength And Weaknesses:**

**Strengths**

- The new augmentation-robust contrastive learning algorithm is an interesting variant of contrastive learning that goes beyond “averaging over augmentations”. To my knowledge this is a novel idea and it also clearly learns better representations than standard contrastive learning, based on the experiments on ImageNet.
- The paper is clearly written for the most part and easy to follow
- Theoretical results are provided to justify the need for going beyond the average over augmentations setting, and also to upper bound the transfer risk for ArCL

**Weakness**

- One of the main concerns about this paper is the use of the term *OOD generalization* for the settings discussed in the paper. In the entire discourse, including experimental evaluations, a representation is learned using unlabeled data from a source domain (ImageNet) and this is evaluated with linear probe or fine-tuning by using **labeled data** from the target domain (CIFAR, Caltech101, etc.). While I am not an expert in OOD generalization, I believe that the standard OOD setting assumes no targets (and sometimes no inputs) from the target domain. I verified this with researchers who constantly publish OOD generalization papers and also checked some surveys (see Section 2.1.2 in [1]). In fact the setting discussed in this paper is much closer to something like “self-supervised transfer learning”. Even the original SimCLR paper [2] evaluate in this setting; see Table 8. OOD generalization is typically evaluated using benchmarks such as WILDS [3]. In light of this, a lot of the claims in the paper will need to be changed, including the title, or actual OOD evaluations need to be included.

- The abstract makes the following claim: **recent work claims that contrastive learning learns more robust representations than supervised learning, our results suggest that this superiority mainly comes from the data augmentation used, i.e., more data are fed to the model.** However I do not think this is adequately justified in the paper. I could not find any experimental evidence to claim that the superiority of contrastive learning mainly comes from “more data fed to the model”. One way to test this is the following: how well does *supervised learning* with the same augmentations do on OOD? If it does well, then one could believe that augmentations played a huge role. Otherwise it could be that contrastive learning somehow learns better features even with the same augmentations. Either the claim needs to be changed to *augmentations can play a role* (based on Section 3), or some experiments are required to justify *superiority mainly comes from …*.

- Section 3 is used to justify that augmentations play a huge role in the good OOD (actually transfer) performance, but the section seemed a little out of place after reading the full paper because it is not really related to the subsequent ideas. It just seems like an “improved upper bound” for the performance of contrastive learning representations on a downstream task. In fact even the analysis from [4] shows an upper bound on supervised learning risk for the augmentation distribution (which they later convert to guarantees for the input distribution). I think the first equation in Theorem B.3 from the arxiv version of [4] shows this. So I think it is important to both, mention the bound in [4] and justify the section better.


Other comments/questions:
- Could it be that the superior performance of ArCL compared to CL is due to the fact that it gets to see more augmentations in total? If so, some ablation study to isolate this effect could be useful
- Treating augmentations as domains for the results in Section 5  is a little weird, because they are not really domains of interest for any downstream evaluation. How does this idea (and Theorem 5.2) connect to the ImageNet experimental settings?
- Is there any connection to ViewMaker network [5] that learn augmentations in an adversarial manner? Those augmentations are more robust to common corruptions in CIFAR. It would be a useful experiment to see if ArCL helps over those augmentations as well.
- Proposition 5.1 $h_A \in \arg\min$ might be more appropriate
- In Theorem 5.2, it seems like the bound if non-vacuous only if $m$ (number of views used) is of the order of $1/c\_{\pi}$. However $c\_{\pi}$ will be at least $|\mathcal{A}|^{-1}$ and so one would need $m$ to be as large as the total number of distinct transformations, which is unreasonably high. Of course this is just an upper bound, but it might be useful to discuss this a bit more. Maybe some covering number argument on $\mathcal{A}$ might lead to a tighter bound since many $A$ might be close to each other.


[1] Shen et al. Towards Out-Of-Distribution Generalization: A Survey

[2] Chen et al. A Simple Framework for Contrastive Learning of Visual Representations

[3] Koh et al. WILDS: A Benchmark of in-the-Wild Distribution Shifts

[4] Haochen et al. Provable Guarantees for Self-Supervised Deep Learning with Spectral Contrastive Loss

[5] Tamkin et al. Viewmaker Networks: Learning Views for Unsupervised Representation Learning

**Summary Of The Paper:**

This paper studies the benefit/role of learning representations with contrastive learning for out-of-distribution (OOD) generalization.
It argues that one way contrastive learning could help with covariate shift is through the diversity in the augmentation distribution, even if the input distributions shift a lot.
This is justified with some theory to upper bound the supervised risk of a representation by the contrastive loss and some other “representation diversity” terms, under some clusterability assumptions on augmentations from prior work.
However the paper finds that vanilla contrastive learning is not sufficient to learn *domain invariant representations* and thus might do poorly in OOD settings.

Viewing different augmentation transformations (like random cropping, gray scaling, brightness) as different “domains”, the paper notes that contrastive learning only tries to make representations invariant to these on average over the augmentation distribution.
Instead it proposes an augmentation-robust contrastive learning (ArCL) that encourages representation invariance for the worst case sets of augmentations, i.e. minimizing $\sup\_{A,A’} ||f(A(x) - A’(x)||^2$ instead of $\mathbb{E}\_{A,A’} ||f(A(x)) - f(A’(x))||^2$.
This ArCL alignment term (combined with MoCo) seems to do much better on linear probing and fine-tuning evaluation while transferring from ImageNet -> one of many downstream image classification tasks.

**Summary Of The Review:**

Overall I think the paper makes interesting contributions to study the benefit of contrastive learning in the transfer setting. While the findings and results paper are quite promising, the issues with the incorrect usage of OOD generalization and some other issues about unjustified claims, I believe that the paper could use benefit from round of reviews. Thus for now I would assign a score of reject.

---

> ### Author Response · Authors · 2022-11-16
> **Response to Reviewer QCLS (3/3)**
>
> **Q5.** Treating augmentations as domains for the results in Section 5 is a little weird, because they are not really domains of interest for any downstream evaluation. How does this idea (and Theorem 5.2) connect to the ImageNet experimental settings?
>
> **A5.** The idea that treats augmentations as domains **comes from the invariant learning** approaches in OOD generalization of supervised learning. Most works in supervised OOD generalization aim to learn features that are invariant across different training domains. How to transfer this idea to CL? Note that usually there is **only one training domain** in CL. However, the purpose of CL is to **align different augmentations**, which shares some similarity with **aligning different domains** adopted in OOD generalization. That is why we treat augmentations as domains. From this point of view, **some domain-invariance notions can be applied to CL (Definition 5.1 and Theorem 5.1).**
>
> In some cases, the downstream domain is indeed some augmented domain. For example, if the downstream domain is obtained by turning the source images blue, and the data augmentation includes color changing, then the target domain is an augmented domain.
>
> In general, as you point out, the downstream domain may not be any augmented domain, such as the ImageNet setting. Nevertheless, augmented-domain invariant features could be more distributional robust. Our ArCL takes advantage of this idea and achieves better transferability in our experiments.
>
> ---
>
> **Q6.** Is there any connection to the ViewMaker network [5] that learn augmentations in an adversarial manner?
>
> **A6.** [5] proposes to learn an adversarial augmentation for more robust representations. Our ArCL can **also be viewed as an adversarial method** since selecting the worst pair will let the loss function larger. However, our approaches have essential differences:
> * ViewMaker aims to **train a generator** to produce adversarial augmentations, while our ArCL is to **select the worst pair** from **given augmentations**, and **does not need any training**.
> * The goal of ArCL is to **learn augmentation-invariant features**, while ViewMaker is to obtain more robust features.
> * For image datasets, ViewMaker may be difficult to be trained on large datasets. Our experiments contain both CIFAR10 and ImageNet.
>
> References:
> [5] Tamkin et al. Viewmaker Networks: Learning Views for Unsupervised Representation Learning
>
> ---
>
> **Q7.** Proposition 5.1  $h_A\in\arg\min$  might be more appropriate.
>
> **A7.** Thanks for mentioning it. Indeed, $h_A\in\arg\min$  is more appropriate. We have revised it in the updated version.
>
> ---
>
> **Q8.** In Theorem 5.2, it seems like the bound is non-vacuous only if the number of views used is of the order of $1/c_{\pi}$. So one would need the number of views to be as large as the total number of distinct transformations, which is unreasonably high. It might be useful to discuss this a bit more.
>
> **A8.** Theorem 5.2 considers the setting where transformations follow a **continuous distribution**. Therefore, the constant $1/c_{\pi}$ is not the total number of augmentations, but is **the "volume"** of the transformation set, and **would not be too large** even if the number of augmentations is infinite. Consider a concrete example: the transformations follow a uniform distribution on the unit ball. Then $1/c_{\pi}=1$, and the bound is non-vacuous. We thank for pointing out the potential usefulness of covering number. Indeed, the proof of Theorem 5.2 is essentially **similar to a covering number argument**, since we need to handle the continuity of distribution. To be specific, let $\widehat{A}\_{m}$ be the set of the $m$ randomly sampled augmentation and $A_1^*,A_2^*$ be two augmentations that attain the supremum in ArCL.  Indeed, what we prove is: with high probability, $\exists A_1,A_2\in\widehat{A}\_m$ that fall in small neighborhoods of $A_1^*,A_2^*$. That is why the dimension term $1/d_\mathcal{A}$ appears in the exponent of $m$: the covering number of a unit ball increases exponentially with $d_\mathcal{A}$.
>
>
> ---
>
> Thanks again for your comments and hope our answers could address your concerns. Please let us know if there is more to clarify.

---

> > ### Comment · Reviewer_QCLS · 2022-11-18
> > **Response to authors**
> >
> > Thank you for the detailed responses. I appreciate replacing OOD to transfer throughout the paper. Overall the author responses and the new experiments have addressed most of my concerns. I will be raising my score to at least weak accept. The title and abstract would also have to be updated eventually.
> >
> > Just a couple of follow up comments:
> >
> > - My suggestion with ViewMaker was that ArCL may have a lesser benefit when combined with ViewMaker augmentations, since they are already selected adversarially. While not necessary, it could be useful experiment to highlight the potential limitations of the ArCL approach
> >
> > - I now realize that Theorem 5.2 is even for continuous distributions. The proof of Theorem 5.1 could be written a bit clearer, especially the step that uses the covering number argument; in particular the step just before $\ge 1 - \epsilon$ could use justification.

---

> > > ### Author Response · Authors · 2022-11-19
> > > **Thanks for your suggestions**
> > >
> > > * Thanks for the advice about ViewMaker and the potential limitation of ArCL. We will conduct new experiments to further study this problem.
> > > * To make it clearer, we provide more justifications for the proof of Theorem 5.2 in the revised paper.
> > >
> > > Thanks again for appreciating our responses and for your detailed suggestions!

---

> ### Author Response · Authors · 2022-11-16
> **Response to Reviewer QCLS (2/3)**
>
> **Q4.** Could it be that the superior performance of ArCL compared to CL is due to the fact that it gets to see more augmentations in total? If so, some ablation studies to isolate this effect could be useful.
>
> **A4.** To make the comparison between ArCL and CL fairer, we **add new experiments for the original CL**. For each sample, we also construct $m$ views and use the expectation of the similarity between positive pairs, namely **average alignment loss**(AAL), as the learning objective. The settings and results are shown in the following.
>
> **Experiment Settings:**
>
> - **Loss Function:** For image $x_i$, the normalized features of its $m$ views from the online branch are $z_{i1},z_{i2},\dots,z_{im}$ and the normalized features from the target branch are $z_{i1}',z_{i2}',\dots,z_{im}'$. Our ArCL alignment loss is
> $$L_{ArCL}^{align}=-\sum_i\min_{j\neq k}z^\top_{ij}z'_{ik}/\tau.$$
>
> And the average alignment loss should be
> $$L_{Average}^{align}=-\sum_i avg_{j\neq k}z^\top_{ij}z_{ik}'/\tau=-\sum_i(\sum_{j\neq k}z^\top_{ij}z_{ik}'/\tau)/(m^2-m).$$
> The uniformity loss keeps the same and the total loss is the sum of the alignment loss and the uniformity loss.
>
> - **Hyperparameters and Datasets:** **For SimCLR**, we train the models on CIFAR10 and evaluate them on modified CIFAR10 and CIFAR100 just as our original paper does. We choose the augmentation views to be 4 and the batch size to be 512. **For MoCo**, We conduct experiments on ImageNet, with the 800-epochs-pretrained model as the initializing model. We train the model for 50 epochs and use the same setting as mentioned in our paper.
>
> **Results.**
> - **Results for SimCLR:** We compare the linear evaluation results on modified CIFAR10 and CIFAR100.
>
> ||Method|Aug 1|Aug 2|Aug 3|Aug 4|Aug 5|Original|
> |---|---|---|---|---|---|---|---|
> |CIFAR10|SimCLR|88.62|86.27|88.96|88.56|88.37|88.81|
> ||SimCLR+Average Alignment Loss(views=4)|88.90|86.34|88.72|88.65|88.44|89.97
> ||SimCLR+ArCL(views=4)|**89.97**|**88.06**|**90.48**|**89.91**|**89.59**|**90.20**
> |CIFAR100|SimCLR|52.28|48.09|53.45|52.58|51.53|53.12|
> ||SimCLR+Average Alignment Loss(views=4)|52.34|48.77|53.12|52.44|51.82|53.21|
> ||SimCLR+ArCL(views=4)|**53.40**|**50.16**|**54.92**|**53.77**|**52.61**|**54.20**
>
>
>
>
> - **Results for MoCo:** We compare the linear evaluation results and finetuning results.
>
> |||ImageNet|Aircraft|Caltech101|Cars|CIFAR10|CIFAR100|DTD|Flowers|Food|Pets|**Avg**
> |:---:|:---------:|:-----:|:-----:|:-----:|:-----:|:------:|:------:|:-----------------:|:--------:|:------:|:------:|:-----:|
> |Linear|MoCo|70.68|41.79|87.92|39.31|92.28|74.90|73.88|90.07|68.95|83.30|72.49|
> ||MoCo+Average Alignment Loss(views=2)|**71.12**|40.53|87.80|38.64|92.23|75.14|**74.95**|88.64|69.24|83.17|72.26
> ||MoCo+ArCL(views=2)|69.70|**44.29**|**89.79**|**42.15**|**93.07**|**76.70**|74.20|**90.40**|**70.94**|**83.68**|**73.91**
> ||MoCo+Average Alignment Loss(views=3)|**71.37**|40.41|87.79|42.09|92.64|75.31|74.89|89.23|69.37|83.79|72.84
> ||MoCo+ArCL(views=3)|69.80|**44.57**|**89.48**|**42.11**|**93.29**|**77.33**|**74.63**|**91.13**|**71.16**|**84.23**|**74.21**
> |Finetune|MoCo||83.56|82.54|85.09|95.89|71.81|69.95|95.26|76.81|88.83|83.30
> ||MoCo+Average Alignment Loss(views=2)||83.87|82.76|85.90|**96.38**|71.43|**72.71**|95.50|76.95|**89.05**|83.84
> ||MoCo+ArCL(views=2)||**86.05**|**87.38**|**87.28**|96.33|**79.39**|72.18|**95.89**|**81.36**|89.03|**86.10**|
> ||MoCo+Average Alignment Loss(views=3)||83.07|83.21|85.19|96.37|72.02|72.55|95.74|79.62|88.83|84.07|
> ||MoCo+ArCL(views=3)||**84.03**|**87.64**|**86.34**|**96.88**|**80.98**|**72.87**|**96.14**|**81.90**|**89.20**|**86.22**
>
> - As the experiments show, when contrastive learning methods with average alignment loss use the same amount of data as ArCL, although they can have higher accuracy compared to vanilla methods (those which only use two views) on ID set, they still have a similar performance with vanilla methods on OOD set, which is much worse than ArCL. This further verifies the superiority of our method.  These experiments are also shown in Appendix E.
>
> ---

---

> ### Author Response · Authors · 2022-11-16
> **Response to Reviewer QCLS (1/3)**
>
> We thank Reviewer QCLS for careful reading and detailed reviews. Indeed, as you have pointed out, the term OOD generalization we have used in our passage is inaccurate. We have revised our paper accordingly. Below, we summarize and address your main concerns.
>
> ---
>
> **Q1.** The term *OOD generalization* is not accurately used. A lot of claims in the paper will need to be changed.
>
> **A1.** Thanks for this detailed and constructive suggestion. We agree with this comment and apologize for our oversight. The term OOD generalization refers to the setting that a model is trained on source domains without access to target data, and then is tested directly on the target domain **(train on sources + test on target)**. In a self-supervised learning regime, this corresponds to pre-train the encoder using unlabeled source data, fine-tuning it using labeled (but still) source data, and then testing the full model on target data **(pre-train on sources + fine-tune on sources + test on target)**. The setting studied by our paper is **pre-train on sources + fine-tune on target + test on target**. Therefore, the term OOD generalization is inappropriate. What we study is indeed the transferability of CL-trained encoders under distribution shift. Therefore, we **replace almost all of the OOD terms with transferability** and modify a lot of claims in our revised paper.
>
>
> ---
>
> **Q2.**  The claim "The superiority of contrastive learning mainly comes from more data fed to the model" in the abstract is not adequately justified.
>
> **A2.** We apologize for this misstatement. What we want to claim is that data augmentation is crucial to the transferability: **more diverse augmentation leads to better transferability.** This is concluded from our Theorem 3.2 and the discussion following it. We have removed these inappropriate statements in the abstract in our revised paper.
>
> ---
>
> **Q3.** Section 3 seems a little out of place. I think it is important to both mention the bound in [4] and justify the section better.
>
> **A3.** Thanks for this suggestion. We have replaced the term OOD generalization with transferability and revised some claims in Section 3. We summarize the contributions of Section 3 in the following.
> - The goal of Section 3 is to study the transferability of contrastive learning, i.e., **on which downstream distributions** the encoder obtained by contrastive learning **has good performance**. By establishing a relationship between supervised loss and contrastive loss, we find the **crucial role of data augmentation** in transferability: contrastive learning performs well on downstream distributions close to the **augmented distribution $\mathcal{D}_{\pi}$**, instead of the original training distribution $\mathcal{D}$.
>
> Thanks for reminding us of a related work [4]. As you point out, Theorem B.3 from the arxiv version of [4] also showed an upper bound for the downstream risk on augmented distribution. We have cited and discussed [4] in our revised paper.
>
> Nevertheless, we note that our paper and [4] take **different technical approaches** and have **different objectives.**
> - [4] studied contrastive learning through the augmentation graph, while ours uses the $(\sigma,\delta)$-augmentation, which is **specifically designed to study the role of data augmentation** in contrastive learning.
>
> - [4] introduced a new spectral loss and established guarantees **only for their proposed objective,** while ours establishes guarantees **for the widely adopted InfoNCE loss,** making it more applicable to explain a range of representative CL methods, e.g. SimCLR, MoCo and many of their variants.
>
> References:
> [4] Haochen et al. Provable Guarantees for Self-Supervised Deep Learning with Spectral Contrastive Loss.
>
> ---

---

### Decision · Program_Chairs · 2023-01-20

**Decision:**

Accept: poster

**Justification For Why Not Higher Score:**

Strong theoretical assumptions and lack of enough motivation for some of them.

**Justification For Why Not Lower Score:**

Lots of interesting theoretical results and good experimental performance for an important problem.

**Metareview: Summary, Strengths And Weaknesses:**

Authors study the effect of augmentations in transferability of self-supervised contrastively-learned encoders for downstream task, a very relevant question. Through a theoretical analysis the paper argues that the augmentation of positive samples leads to better downstream performance on datasets with similar distribution as augmented version of original pre-training dataset. Some of the strong assumptions used aren't motivated enough and their necessity is not proven. This theory also assumes that the conditional label distribution for downstream tasks don't change downstream, and peculiarly ignores any effect of negative samples. It is implied that augmentations may be the reason contrastive pre-training transfers in practice, however this totally ignores the effect of negative samples. Though the theory is not too surprising and it is done in population level, it could still be a useful guide for practioners. I am hoping the authors would add the discussion requested by reviewers in the main text.

Next authors prove that adversarially (instead of randomly) choosing augmentations from a class of augmentations for positive samples can improve "adverarially" chosen downstream task's performance. Motivated by this observation, they propose ArCL method which does exactly same and this method improves transfer performance on many models, augmentations, and datasets. This concept is very similar to "hard-positive mining" from [1], where instead of searching in space of real samples ArCL searches in the space of augmented samples. ArCL seems powerful in practice and is conceptually simple so that it can be added to any augmentation-based pre-training task, although this could increase computational complexity.

Nit: mu_k is defined without using the subscript k.

[1] Schroff et al., FaceNet: A Unified Embedding for Face Recognition and Clustering, https://arxiv.org/pdf/1503.03832.pdf

**Note From Pc:**

if the above contains the word "oral" or "spotlight" please see: "oral" presentation means -> notable-top-5% and "spotlight" means -> notable-top-25%. As stated in our emails, we are disassociating presentation type from AC recommendations